# Unsupervised fake news detection on social media using hybrid Gaussian Mixture Model

Sajida Perveen[1], Muhammad Shahbaz[2], Sami S. Albouq[3], Khlood Shinan[4], Hanan E. Alhazmi[5], Fatmah Alanazi[6], M. Usman Ashraf[7]*, Rehan Ashraf[1]

1 Department of computer Science, National Textile University, Faisalabad, Pakistan, 2 Department of Computer Engineering, University of Engineering & Technology, Lahore, Pakistan, 3 Faculty of Computer and Information Systems, Islamic University of Madinah, Madinah, Saudi Arabia, 4 Department of Computers, College of Engineering and Computers in Al-Lith, Umm Al-Qura University, Makkah, Saudi Arabia, 5 Department of Cybersecurity, College of Computing, Umm Al-Qura University, Makkah, Saudi Arabia, 6 Computer Science Department, College of Computer and Information Sciences, Imam Muhammad Bin Saud University, Riyadh, Saudi Arabia, 7 Department of Computer Science, GC Women University Sialkot, Pakistan

* usman.ashraf@gcwus.edu.pk

## Abstract

The rise of social media has revolutionized information dissemination, creating new opportunities but also significant challenges. One such challenge is the proliferation of fake news, which undermines the credibility of journalism and contributes to societal unrest. Manually identifying fake news is impractical due to the vast volume of content, prompting the development of automated systems for fake news detection. This challenge has motivated numerous research efforts aimed at developing automated systems for fake news detection. However, most of these approaches rely on supervised learning, which requires significant time and effort to construct labeled datasets. While there have been a few attempts to develop unsupervised methods for fake news detection, their reported accuracy results thereof remain unsatisfactory. This research proposes an unsupervised approach using clustering algorithms, including Gaussian Mixture Model (GMM), K-means, and K-medoids, to eliminate the need for manual labeling in detecting fake news. In particular, it also proposes a novel hybrid method that leverages the Gaussian Mixture Model (GMM) in conjunction with the Group Counseling Optimizer (GCO), a metaheuristic optimization algorithm, to identify the optimal number of clusters for the detection of fake news. The comparative analysis of the evaluation results on real-world data demonstrated that the proposed hybrid GMM outperforms the state-of-the-art techniques, with a silhouette score of 0.77, ARI of 0.83, and a purity score of 0.88, indicating a significantly improved quality of clustering results.

**Data availability statement:** https://github.com/Sajida-Perveen/Project-code.

**Funding:** The author(s) received no specific funding for this work.

**Competing interests:** The authors have declared that no competing interests exist.

## 1 Introduction

The emergence of the Internet and the increasing popularity of social media platforms, such as Facebook and Twitter, have revolutionized the way information is disseminated in today's world. [1,2]. This revolution has led news providers to transition from traditional media formats, such as newspapers and magazines, towards digital formats including online news platforms, blogs, and social media feeds, enabling them to deliver up-to-date news in near real-time to their subscribers [3,4]. As the internet usage has expanded, an increasing number of consumers turned away from traditional media channels, preferring to access information through online platforms [5,6]. This shift has made it incredibly convenient for consumers to access the latest news instantly [7,8]. Consequently, web portals have become a popular tool for delivering and receiving news, especially for users who are always on the move [9]. Notably, Facebook alone drives 70% of the traffic on news websites [10].

Social media platforms have become powerful tools for discussion, idea-sharing, and debate on topics like democracy, education, and health [11–14], enabling decentralized content creation and consumption [15]. However, alongside these benefits come significant challenges [2,11,16–19]. The overwhelming volume of information can make it difficult to distinguish credible sources [19,20]. Moreover, these platforms are often misused for financial gain, spreading biased views, manipulating opinions, and disseminating misinformation or fake news [21–24].

The phenomenon of fake news is not a recent development. It existed even long before the advent of the Internet [25]. Allcott and Gentzkow [26] initially defined the concept of fake news as news that is intentionally fabricated and verifiably false and designed to mislead readers [27]. Another widely used definition of fake news is that fake news itself is not actual but is made real for a specific purpose [28]. Additionally, the dissemination of fake news on social media platforms is exacerbated by its user-friendly and rapid sharing capabilities, which provide remarkable convenience in the creation and distribution of misinformation, further imploring the audience to disseminate the content at an exponential pace on the Internet. Unlike social media, mainstream media follow editorial standards and guidelines, and are operated by professional journalists [29]. On the other hand, social media primarily relies on user-generated content with limited editorial controls or gatekeeping procedures [30]. Instead of encountering reliable and unbiased information, users are inundated with a multitude of fake news on social media platforms [31,32]. Therefore, social media has also become a breeding ground for misinformation [33].

There have been numerous examples of fake news throughout history. A recent example is the proliferation of information about COVID-19. In late 2019, massive amounts of informative data related to COVID-19 pandemic were generated and disseminated on social media networks. Although, these networks make it possible to disseminate information to a large audience [34]. Unfortunately, not all disseminated information about the outbreak is accurate and trustworthy. During this era, some of the information disseminated around these networks was identified as fake news or even exacerbated the crisis by promoting unproven treatments and fueling vaccine

hesitancy leading to detrimental health, social, and cultural consequences [35]. Thus, WHO called it an "information epidemic". Fake news can also incite violence, panic, or harm to public safety in various contexts [36,37]. Unfortunately, the epidemic spread of fake news is a byproduct of the popularity of the social media platforms. According to a report, only 26% of the participants expressed a high level of confidence in discriminating between real and fake [28,38]. This ratio is comparatively low and indicates insufficient capacity to discriminate between genuine and fake news.

Consequently, it has become increasingly challenging for authoritative institutions to effectively combat the rapid dissemination of fake news. Although significant efforts have been made to address this issue, researchers are increasingly turning to artificial intelligence (AI) solutions [39]. Machine learning, particularly natural language processing techniques, have shown promising contribution in fake news detection [40–43]. Other researchers [44–46] explored to utilize a combination of different deep learning models. Some have even demonstrated that deep learning techniques outperform traditional machine-learning techniques [46]. However, the inherent "black-box" nature of advanced deep learning models presents significant challenges, particularly in fields requiring transparency and interpretability [47].

According to the existing literature, considerable attention has been given to machine learning-based supervised methods for detecting fake news [41–43]. These studies developed classification models using different sets of features including user profiles, news content [29], message propagation [48] and social contexts [23,49–51]. While these models have shown promising results to some extent, these supervised methods require labeled data to build a classification model [52,53].

However, obtaining a large number of annotations is time consuming and labor intensive, as the process requires careful examination of news content, as well as other additional evidence such as authoritative reports [54,55]. Leveraging a crowdsourcing approach to obtain annotations could alleviate the burden of expert checking; however, the quality of annotations may be compromised [56]. As fake news is intentionally written to mislead readers [57], individual human workers alone may not possess the domain expertise to differentiate between real and fake news [58]. Besides this, fake news detection often involves identifying and tracking patterns in data that change dynamically over time, such as shifts in the spread of misinformation, the emergence of new narratives, or changes in the behavior of online communities. These evolving patterns require models that can adapt to and account for changes in the data distribution, similar to how dynamic systems evolve based on state variables and external inputs.

In the context of Gaussian Mixture Models (GMMs) [59], these models have been widely applied for modeling dynamic systems to capture the underlying probabilistic structure of time-varying processes [60–67]. Similarly, in fake news detection, GMMs can be used to model the distribution of news articles or social media posts that change over time, grouping them into clusters that reflect the underlying patterns of authenticity versus misinformation [58]. By leveraging GMMs, we can identify clusters of fake and real news articles that evolve as new information is disseminated, without requiring labeled data for training [68].

Although, GMM is a potentially powerful technique for fake news detection, it has several limitations when applied to textual data [68]. As, it relies on the assumption that data is generated from a mixture of Gaussian distributions, which may not hold for textual data, as these features often exhibit complex, non-Gaussian patterns [69]. Additionally, textual data is typically high-dimensional, and GMM can struggle with the curse of dimensionality [64,70–73]. The Expectation-Maximization (EM) algorithm used in GMM is also sensitive to choice of initial parameters selection, potentially leading to convergence on local optima rather than the global optimum, leading to erroneous results [74–76]. Overall, these limitations highlight the need for additional techniques or enhancements to improve the model's effectiveness in fake news detection. Unlike EM algorithm, GCO, a meta-heuristic optimization technique, explores the parameter space more effectively through iterative improvements based on group counseling behavior, helping find the global optimum and avoiding local optima [77,78]. GCO handles large datasets well, making it suitable for real-world news classification [79]. To address these challenges, this paper presents a novel hybrid Gaussian Mixture model, GCOGMM, a framework to identify fake news in a fully unsupervised manner. The primary objective of this study is to optimize parameters selection

and avoid local optima, thereby enhancing the model's ability to effectively handle complex, high-dimensional textual data to effectively identify fake information.

The following are the key aspects of this paper:

1. The proposed research integrates Group Counseling Optimization (GCO) with the Gaussian Mixture Model (GMM) to optimize parameters selection and avoid local optima, thereby enhancing the model's ability to effectively handle complex, high-dimensional textual data.

2. To explore the automatic fake news detection based on the analysis of surface-level linguistic patterns.

3. A comprehensive evaluation analysis of the proposed approach is proformed on benchmark dataset. Particularly, for evaluation, the proposed research developed three baseline unsupervised methods for detecting fake news, namely k-means, K-mediods and Gaussian Mixture Models (GMM).

4. Additionally, to the best of available knowledge, the utility of the proposed approach when combined with standard GMM has not been investigated to address the issue of fake news identification on social media, which has become a daunting societal and individual issue. Furthermore, evaluation results depicted that the proposed method outperforms over all the baseline models. It also demonstrated that this hybrid approach can improve a text-only Gaussian model.

## 2 Proposed methodology

### 2.1 Data representation

In this study, we aimed at to enhance the current state of the art in fake news detection, with the ultimate goal of developing reliable and robust tools to combat misinformation in the digital era. To achieve this, we utilized the publicly available LIAR dataset for fake news detection, collected from POLITIFACT.COM. This dataset comprises 12.8K short statements related to various contexts collected over a decade. After data acquisition, the next crucial step is data processing. These preprocessing steps are vital for cleaning and standardizing such textual data, making it more suitable for natural language processing (NLP) tasks.'

The dataset provides a detailed classification of news articles into six subcategories: True, Mostly True, Half True, Barely True, False, and Pants on Fire. These subcategories offer a nuanced perspective on the varying degrees of truthfulness within the dataset. These subcategories are represented by varying frequencies within the dataset: Half True (2,114 instances), False (1,994 instances), Mostly True (1,962 instances), True (1,676 instances), Barely True (1,654 instances), and Pants on Fire (839 instances). Each data point in the dataset have a unique ID. This distribution reveals an imbalance in the dataset, with certain categories like Half True and False being more prevalent, while others such as Pants on Fire are significantly less represented. This imbalance highlights the complexities of truthfulness identification and may pose significant challenges during the analysis process. To address this problem and ensure an equitable representation across all the categories, we balanced the dataset by randomly selecting 839 instances from each category without replacement. This approach helps to create a more uniform dataset. However, to ensure that the clusters formed reflect the actual structure of the data, rather than being influenced by human-imposed labels that may not fully capture the complexity or nuances of the dataset, all pre-existing labels are removed before conducting the clustering analysis. This approach aligns with the principles of unsupervised learning and is essential for achieving accurate, data-driven clustering outcomes in the context of this study.

### 2.2 Preprocessing

**2.2.1 Converting all data to lower case.** As a written document is composed of sentences, and sentences can comprise a combination of uppercase and lowercase letters, the initial preprocessing step involves converting all text data to a single text case. We have chosen lowercase for this purpose. This is crucial to ensure consistency in the text analysis

procedure [80,81]. By converting everything to lowercase, we eliminate potential inconsistencies that may arise from various cases of the same word.

**2.2.2 Tokenization.** It is a crucial step in NLP where a stream of text is divided into smaller units, which typically referred to as tokens [82]. These tokens can be words, phrases, symbols, or individual characters, depending on the level of granularity needed for the analysis. This process is essential for many NLP tasks as it facilitates the understanding and manipulation of text by converting it into a format that computational models can process. Thus tokenization is often the first step in a pipeline that includes other tasks, which collectively enhance the performance of NLP applications [82]. Therefore, tokenization is performed on each sentence in the dataset.

**2.2.3 Removing non-alphanumeric characters.** Following tokenization, it is quite common to remove non-alphanumeric characters from the text. This particular step includes eliminating symbols, punctuation marks, and special characters that do not hold substantial semantic meaning in the text [83]. Examples of non-alphanumeric characters include commas, periods, exclamation marks, and other symbols that are not part of words or numbers.

**2.2.4 Removing stop words and punctuation.** Stop words are indeed terms that are commonly found in documents and are considered to have little semantic meaning in the context of text analysis. Their removal is a standard preprocessing step in NLP and text mining applications to improve the performance and efficiency of these systems [84]. Therefore removed from the text during the analysis process. Examples of stop words include "the," "is," "and," "of," and so on. The removal of stop words helps to reduce noise in the text data and concentrate on more significant terms. Moreover, the elimination of punctuation marks further refines the text and prepares it for analysis. NLTK library in Python contains a list of stop words for the English language.

**2.2.5 Stemming.** Stemming is a process widely used to reduce words to their derived words known as stem, base, or root form. This process typically entails the elimination of extraneous characters to normalize words, which involves removing prefixes and suffixes [85,86]. For instance, the words "write", "wrote,", "written" and "writing" would all be stemmed to the base form "write" By stemming words, the vocabulary size is reduced, and similar words are consolidated, which can enhance the effectiveness of text analysis algorithms [87]. Porter Stemmer is one of the stemming model which is used in this study to convert the words into their root form.

**2.2.6 Creating word cloud.** After applying the aforementioned preprocessing steps, a high-dimensional feature space of 184,412 unique tokens were obtained and are reunited to form a processed text dataset that is prepared for further analysis. This consolidated text preserves the pertinent semantic information while eradicating extraneous details and noise, making it appropriate for tasks such as text classification, sentiment analysis, or information retrieval. Fig 1 showcases a more detailed representation of the dataset in the form of word cloud. The conspicuous presence of political news and the substantial inclusion of global events indicate particular thematic priorities within the media. This observation invites a deeper analysis of the content, aiming to uncover potential underlying influences that may be shaping the portrayal of political and global events in the dataset.

## 2.3 Feature extraction or vectorization

**2.3.1 Text vectorization using term frequencey-inverse document freuency.** The choice of metrics for encoding textual data is pivotal in determining the performance and suitability of a particular model. While advanced encoding methods like contextual embeddings or topic modeling techniques (e.g., BERT, LDA) offer deeper semantic understanding, their high computational cost and preprocessing requirements [88] may not align with the study's objectives. As the primary research objective is to enhance the efficacy of detecting fake news using unsupervised techniques. Therefore, to achieve this, TF-IDF (Term Frequency-Inverse Document Frequency) [89] was chosen based on its balance between simplicity, interpretability, and its effectiveness in identifying informative features within textual data. It excels in distinguishing relevant terms by penalizing high-frequency words that are common across documents, making it suitable for clustering tasks where capturing term-specific relevance is crucial.

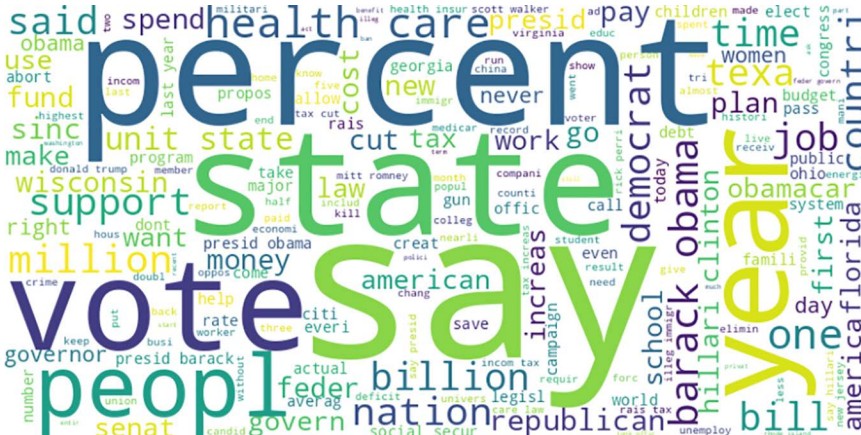

**Fig 1. Abstract representation of the dataset in the form of word cloud.**

Vectorizer as a critical element in the preprocessing of textual news data. The TF-IDF Vectorizer is a widely adopted technique in natural language processing, taking into account both the frequency of terms in individual documents and their significance across the entire corpus [88,90]. The vectorization process essentially entails assigning numerical values to words based on their occurrence frequency in a document (Term Frequency) and inversely scaling these values by their frequency in the entire corpus (Inverse Document Frequency), thereby emphasizing terms that are both frequent in a document and unique across the corpus. Mathematically, the TF-IDF can be expressed as:

$$tf-idf_{t,D} = TF_{t,D} * IDF_t \tag{1}$$

$$TF_{t,D} = \frac{\text{Number Of Repetitions of Term } t \text{ in a Document D}}{\text{Total No. of terms n a Document}} \tag{2}$$

And

$$IDF_t = \log \frac{\text{No. of Documents}}{\text{Number Of Documents Containing The term } t} \tag{3}$$

Thus, the TF-IDF is instrumental in transforming unstructured textual information into a format that is compatible with machine learning techniques [91]. It represents text as numerical vectors enables algorithms to operate on the data efficiently to extract meaningful patterns, and make informed decisions.

**2.3.2 Principal Component Analysis (PCA).** After carrying out above mentioned preprocessing tasks, it is necessary to decrease the number of features to lower computational complexity and improve performance [92–94]. The optimal selection of feature extraction algorithm is crucial to the quality of the results [95]. According to existing literature to deal with the high dimensional data Information Gain (IG) [96], Mutual Information [97,98], Gini Coefficient (GI), Term Frequency-Inverse Document Frequency (TF-IDF), Principal Component Analysis (PCA) [27,99,100] and Chi-Square Statistics (CHI) [101] are frequently used for feature extraction. However, to improve the scalability of the obtained results, PCA is widely used as the dimensionality reduction approach in fake news detection applications [102–106].

PCA applies a linear transformation to reduce the dimensionality of a feature set, simplifying the dataset while preserving the essential characteristics of the original data [102]. The transformed dataset may have the same or fewer features

compared to the original. Principal components are derived by calculating the covariance matrix [107]. These components are arranged in decreasing order of importance.

Let X denote the Dataset with n observations and p variables and $\Sigma$ represent the covariance matrix of mean centered dataset $\acute{X}$. $\Sigma$ matrix captures the pairwise covariance between variables as depicted below:

$$\Sigma = \frac{1}{n-1} \sum_{i=1}^{n} (\acute{X}_i - \overline{X})^T (\acute{X}_i - \overline{X})$$

(4)

To find principal components eigenvalue decomposition of $\Sigma$ matrix is performed. Subsequently the eigenvectors corresponding to the largest eigenvalues were selected as principal components as depicted below:

$$\Sigma = Q \bigwedge Q^T$$

(5)

Q is the matrix of eigenvectors of $\Sigma$ derived from the text vectors, where each column represents an eigenvector. Where $\Lambda$ is a diagonal matrix containing the corresponding eigenvalues.

This reduction in dimensionality facilitated faster processing and aided in uncovering underlying patterns and structures within the textual data. The transformed data, now represented in a lower-dimensional feature space, served as the input for our clustering algorithm as depicted in Fig 2.

**2.3.3 Gaussian mixture model.** A Gaussian Mixture Model (GMM) is an unsupervised probabilistic model that represents a dataset as a mixture of several Gaussian distributions [108]. GMM is parameterized by two type of values, the component weight and the component mean and variance. For a GMM with k Gaussian component, each K in the mixture model represents a cluster or subgroup within the data. GMM serves as an effective means for addressing intricate data distributions that cannot be easily segregated using conventional clustering techniques. In a multivariate GMM, each K is characterized by its mean $\mu_K$ Covariance matrix $\Sigma_K$ and weight $\pi_K$ the probability of belonging to component k. The key components of a multivariate GMM include the Probability Density Function (PDF) of a multivariate Gaussian distribution, the mixture model PDF, and the Expectation-Maximization (EM) algorithm for parameter estimation. The probability density function of a multivariate Gaussian distribution with mean vector and covariance matrix is calculated as: below:

$$\mathcal{N}(x|\mu, \Sigma) = \frac{1}{(2\pi)^{\frac{D}{2}} |\Sigma|^{1/2}} \exp\left(-\frac{1}{2}(x-\mu)^T \Sigma^{-1}(x-\mu)\right)$$

(6)

Where x is a D-dimensional data point μ is the D-dimensional mean vector, $\Sigma$ is a D×D variance matrix, $|\Sigma|$ determinant of the variance matrix and $\Sigma^{-1}$ is the inverse of the variance matrix, allowing the model to capture complex, multidimensional relationships within the data..

The PDF of a multivariate Gaussian Mixture Model (GMM) with K components is given by the weighted sum of the PDFs of its components as depicted below:

$$P(x) = \sum_{k=1}^{K} \pi_k \ \mathcal{N}(x|\mu_k \Sigma_k)$$

(7)

Here P represents the probability of component k such that $\sum_{k=1}^{K} \pi_k = 1$. Whereas $\mu_k$ is the D-dimensional mean vector of component k, $\Sigma_k$ is the variance matrix of component k and $\mathcal{N}(x|\mu_k, \Sigma_k)$ is the multivariate Gaussian PDF of component k. Subsequently, Expectation-Maximization (EM) Algorithm for Multivariate GMM consists of two steps. The first step,

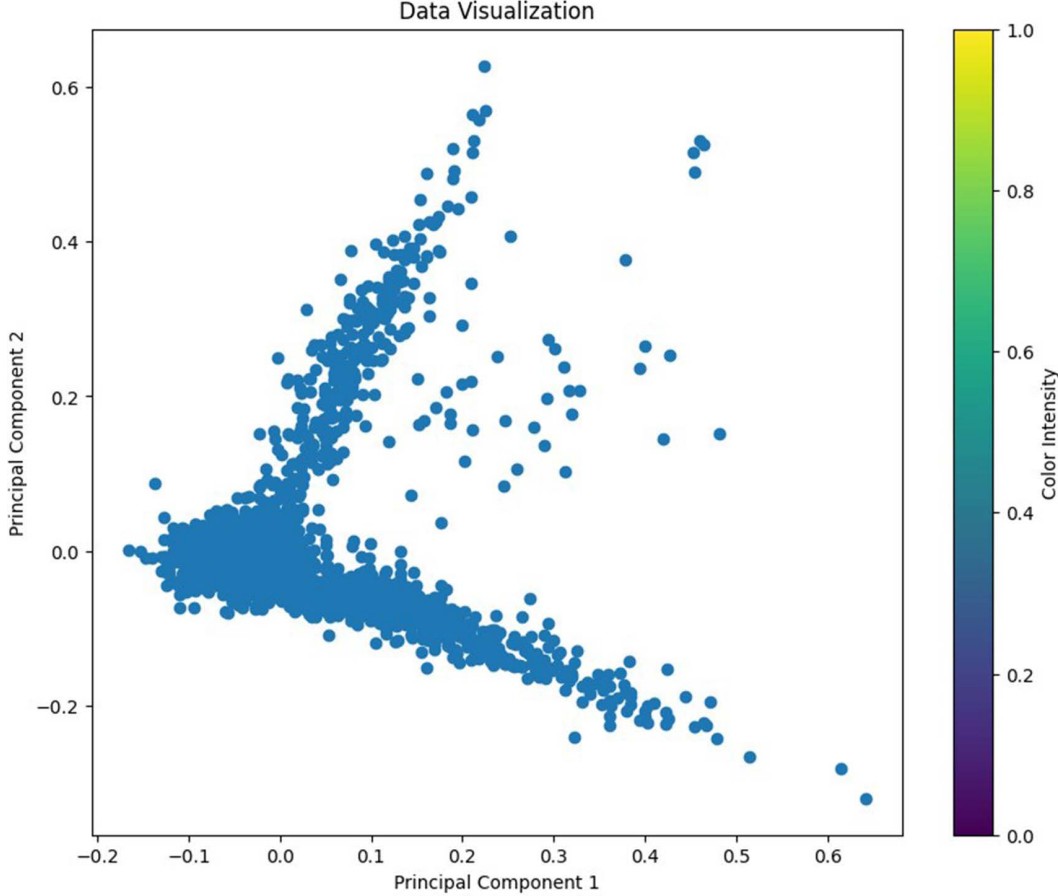

**Fig 2. A visual representation of input data points transformed into a lower-dimensional feature space using PCA.**

known as the expectation step to calculate the probability for each data point $x_i$ to each $C_k$ where $x_i \in X$ given the model parameters $|\mu_k, \Sigma_k$ and $\pi_k$.

$$\gamma(\mathcal{Z}_{nk}) = \frac{\pi_k \mathcal{N}(x_k|\mu_k, \Sigma_k)}{\sum_{j=1}^{K} \pi_j \mathcal{N}(x_j|\mu_j, \Sigma_j)}$$

(8)

$\gamma(\mathcal{Z}_{nk})$ is the probability that data point $x_k$ belongs to component k. whereas the second step known as Maximization in this step, the parameters $\pi_k$, $\mu_k$ and $\Sigma_k$ are updated based on the $\gamma(\mathcal{Z}_{nk})$ computed in the expectation step. Thus, $\gamma(\mathcal{Z}_{nk}) = p(C_k|x_i, \mu, \Sigma, \pi)$ where.

$$p(C_k|x_i, \mu, \Sigma, \pi) = \sum_{i=1}^{n} \log \left( \sum_{k=1}^{K} \pi_k \mathcal{N}(x_i|\mu_k, \Sigma_k) \right)$$

(9)

Here $x_i$ is the data point , $\mu, \Sigma, \pi$ are the parameters of the GMM and $\mathcal{N}(x_i|\mu_k, \Sigma_k)$ is the probability density function of component k evaluated at a data point $x_i$. This process continues iteratively until convergence criteria are met, such as changes in log-likelihood or parameter stability.

## 2.4 Hybrid GMM using Group Counseling Optimizer (GMM_GCO)

Incorporating Group Counseling Optimization (GCO) with a Gaussian Mixture Model (GMM) automates and improves the parameter tuning process addressing the limitation of traditional GMM. As discussed above the performance of traditional GMM is sensitive to key hyper-parameters, such as the number of components and the convergence threshold of the Expectation-Maximization (EM) algorithm. Furthermore, manual tuning methods often fail to ensure optimal performance, especially with complex datasets. Group Counseling Optimization (GCO) is a a population-based metaheuristic optimization algorithm inspired by the idea of collaborative learning and behavioral improvement in group counseling dynamics. It avoids local optima primarily through population diversity, iterative feedback mechanisms, and dynamic exploration-exploitation balancing.

Unlike traditional optimization algorithms that follow a single trajectory (e.g., gradient descent), GCO maintains a diverse population of candidate solutions. It maintains a diverse population of candidate solutions, allowing it to explore multiple regions of the solution space simultaneously and avoid being trapped in local optima. Individuals learn from better-performing peers through guided interactions, helping weaker candidates move toward more promising areas. Furthermore, during each iteration, individuals undergo small perturbations or adjustments that introduce randomness in their behavior (solution state). This allows GCO to explore new areas in the solution space rather than converging prematurely to a suboptimal point. The randomness ensures that even if the population converges temporarily, there's always a chance of discovering better global optima.

It also balances exploitation of strong solutions with exploration of new possibilities, ensuring a robust search process. Over successive iterations, the population improves collectively, enhancing convergence toward globally optimal solutions. It efficiently explores the parameter space, preventing local optima and ensuring globally optimized solutions. This integration not only addresses the limitations of traditional GMM but also provides a scalable and purely unsupervised framework suitable for real-world applications where labeled data is limited or unavailable. The pseudocode of the proposed Hybrid GMM is depicted in Fig 3.

Thus, to optimize a Gaussian Mixture Model (GMM) for fake news detection using Group Counseling Optimization (GCO), we start by initializing the population of agents. An agent represent an individual candidate solution in the search space. Each agent in the population is represented by a parameter vector $x_i = [x_{i1}, x_{i2}, x_{i3}, \ldots\ldots, x_{id}]$, where each parameter $x_{ij}$, is randomly generated within its respective bounds $[l_i, u_i\}$. Here, bounds variable particularly defines the ranges of the parameters that the Group Counseling Optimization (GCO) algorithm will optimize. Each element of the bounds list is a tuple that specifies the lower and upper limits for a particular parameter. Specifically, the bounds cover key GMM hyper-parameters, including the number of components, Regularization parameter for covariance matrices, Maximum number of iterations for the Expectation-Maximization (EM) algorithm, and tolerance (Convergence threshold. Mathematically, each parameter $x_{ij}$, for agent $i$ is drawn from a uniform distribution within its bounds $x_{ij} \sim u(l_i, u_i)$.

In the proposed GCO-based optimization, the fitness of each agent is evaluated using an internal clustering validation metric Silhouette Score rather than external ground truth labels. This choice ensures that the entire optimization process remains fully unsupervised and does not rely on labeled data during training process. The fitness of each agent is then evaluated using an objective function can be defined as:

$$f(x) = -\text{Silhouette Score}(x) \tag{10}$$

This transformation is used because optimization algorithms, including GCO, are typically designed to minimize an objective function.

The optimization process involves iterating for a predefined number of iterations. In each iteration, the mean vector of the population is calculated, and a random vector with values in the range [− 1] is generated. Each agent's parameters are updated based on the difference between the agent's parameters and the mean vector, scaled by the random vector

**Hybrid GMM with Group Counseling Optimizer (GMM_GCO)**

1: Input: Given dataset X={ x_1,x_2,x_3,......,x_(n )}

   //{No. of Components, Regularization Covariance, Max_iteration, tolerance} ∈ bounds

2: Randomly initialize the bounds,num_agents,

3: Generate num_agents random solutions within the bounds

$$Let\ x = [n_{components}, reg_{covar}, max_{iter}, tol]$$

4: Given a parameter vector x, compute the objective function as follows:

$$f(x) = arg\ \min_x \left( \frac{1}{N} \sum_{i=1}^{N} \mathbb{1}(y_i = \hat{y}_i(x)) \right)$$

Step 5: Group Counseling Optimization (GCO) Process

   Procedure Group_Counseling_Optimization ($f(x)$, $bounds$, $num\_agents$, $max\_iter$ )

   Compute Initial Fitness f($x_i$) for each solution in the X

// $x_i$ is the parameter vector for the ith agent within bounds $[l_i, u_i]$ and $x_i = [x_{i1}, x_{i2}, x_{i3}, ..., x_{id}]$

   For t=1: Max_Iterations

      For each agent i in X

         $M = \frac{1}{|X|} \sum_{j=1}^{X} x_i$  // calculate the population mean and X is the number of agents

         $x_{new} = x_i + r \odot (x_i - M)$     //new Candidate solution

         $x_{new} = max(min(x_{new}, u)\ l)$   // check bounds $[x_{new} \sim u(l_i, u_i)]$

         $f_{new} = f(x_{new})$  // Evaluate Fitness

         if $f_{new} < f(x_i)$: $x_i = x_{new}$ $f(x_i) = f_{new}$   // update the agent's parameters

         $x^* = arg\ \min_{x_i} f(x_i)$

      Repeat until convergence or maximum iterations.

   Return the best parameters $x^*$ with the lowest fitness.

6: Initialize the GMM with the optimized parameters ($x^*$) found by GCO.

   GMM=GaussianMixture ($n_{components} = round(x_1), reg_{covar} = x_2, max_{iter} = x_3, tol = x_4]$ )

   // where$x_1, x_2, x_3, x_4$ are the components of $x^*$

   GMM.fit(Scaled_X)    // Model training

   GMM.predict(X)

**Fig 3. Pseudocode for Hybrid GMM and Pseudocode of proposed Hybrid GMM.**

$r$. The updated parameters are adjusted to ensure they range within the predefined bounds. The fitness of the updated agent $f(x_{new})$ is then evaluated. If the new fitness is better (lower) than the current fitness, the agent's parameters are updated to the new values. This process is repeated for all agents in the population and continues until convergence or maximum iterations are reached.

After completing the iterations, the agent with the best (lowest) fitness in the population is identified as the best solution $x^* = arg\ min f(x_i)$. This optimization process using GCO ensures that the GMM parameters are tuned to achieve the highest accuracy possible to effectively distinguish between real and fake news.

## 2.5 Classical clustering techniques

### 2.5.1 K means clustering.
K-means clustering is a fundamental clustering algorithm within the realms of unsupervised techniques. It discover concealed patterns within the data while effectively grouping data points that share similarities together [109]. Its primary objective is to iteratively partition the data points into k non overlapping clusters. During this partitioning process, each cluster finds its representation through a centroid point based on a distance metric [110]. This process aims to minimize intra-cluster variance and maximize inter-cluster variance, leading to well-defined clusters [111].

Let D is a D-dimensional dataset of n observations represented as $X = \{x_1, x_2 \ldots \ldots x_n\}$ where each data point belongs to a cluster C = {$C_1$, $C_2$,....., $C_k$} based on some proximity measure. In this research to find out the similarity between data points we incorporated Euclidean distance as a proximity measure or partitioning method Dist(i, j) that assures how similar or distant observation i from j are; consequently build k distinct partitioning from a D dataset of n observations. Where each k corresponds to a distinct cluster consist of at least one object and K ≤ n. Euclidean proximity measure can be determined as follows.

$$arg\ min \sum_{i=i}^{k} \sum_{x \in Ci} ||x - \mu_i||^2 = arg\ min \sum_{i=1}^{k} |C_i|\ Var\ C_i$$

(11)

Where $\mu_i$ represents the centroid of data points in $C_i$ can be calculated as:

$$\mu_i = \frac{1}{|C_i|} \sum_{x \in Ci} x$$

(12)

In this research, we also incorporated K-means clustering to group TF-IDF vectors of news data with similar characteristics, aiding in the identification of patterns and clusters within the dataset. However, it requires specifying the number of clusters (K) in advance, which is a challenging task. In this research we incorporated elbow and Bayesian Information Criterion (BIC) techniques to address this challenge.

### 2.5.2 K-medoids clustering.
K-medoids clustering [112] is similar to K-Means, but instead of using the mean of the data points to determine the cluster centre point, it uses actual data points as the cluster centers (medoids). For each data point $x_i$ find the medoid $m_j$ that best represents the center of each cluster by minimizing the sum of distances within each cluster, enhancing the robustness and interpretability of the clustering results.

Let $M = \{m_1, m_2 \ldots \ldots m_k\}$ be the set of randomly selected initial medoids from the dataset $X$
where $X = \{x_1, x_2 \ldots \ldots x_n\}$. For each data point $x_i$, find the medoid $m_j$, that minimizes the distance:

$$Assign\ (x_i) = arg\ \min_{x \in c_j} d(x_i, m_j)$$

(13)

Mathematically, for each cluster $c_j$ with data points $\{x_{j1}, x_{j2} \ldots \ldots x_{jn}\}$, the new medoid $m_j$ is selected as:

$$Assign\ (x_i) = arg\ \min_{x \in c_j} \sum_{x_i \in c_j} d(x, x_i)$$

(14)

## 2.6 Determine the optimal number of components for clustering algorithms

### 2.6.1 Elbow method for K-means and K-medoid Clustering.
The Elbow method is a commonly used technique for identifying the optimal number of clusters in a dataset when applying clustering algorithms like K-means [113] and K-medoids [114]. This method relies on the principle that the within-cluster sum of squares (WCSS) typically decreases as the number of clusters increases. Mathematically, WCSS can be defined as:

$$WCSS(k) = \sum_{i=i}^{k} \sum_{x \in Ci} ||x - \mu_i||^2$$

(15)

Where k is the number of clusters, x is the data point in the ith cluster with $\mu_i$ centre point.

The Elbow method involves running the algorithm for varying numbers of clusters and plotting the corresponding WCSS values. The resulting plot often resembles an "elbow" shape, where the WCSS decreases rapidly initially and then starts to level off. The optimal number of clusters is identified at the "elbow point," which is where the rate of decrease in WCSS ($\Delta$WCSS (k)) starts diminishing considerably [56]. This point indicates a balance between minimizing WCSS (which reflects cluster compactness) and avoiding over-fitting by using too many clusters [115]. This involves calculating the difference in WCSS for consecutive values of k as depicted below:

$$\Delta WCSS(k) = WCSS(k-1) - WCSS(k)$$

(16)

**2.6.2 Bayesian Information Criterion (BIC) for GMM.** In this study, we employed the Bayesian Information Criterion (BIC) [116] to determine the optimal number of components for a Gaussian Mixture Model (GMM) applied to our dataset. The BIC is a statistical criterion used for model selection, balancing model complexity and goodness of fit [117].

We considered a range of candidate numbers of components and explored different covariance types. Then for each combination of parameters, we trained GMM ($\mathcal{M}_k$). We computed the BIC for each trained model $\mathcal{M}_k$ as depicted below:

$$BIC\left(\mathcal{M}_k\right) = -2 \ln\left(\acute{L}\left(\mathcal{M}_k\right)\right) + p_K \ln(N)$$

(17)

Where $\acute{L}\left(\mathcal{M}_k\right)$ is the likelihood of the data under the model $\mathcal{M}_k$ and $p_K$ is the number of parameters in $\mathcal{M}_k$. $\mathcal{M}^*$ is the model with lowest BIC as depicted below:

$$\mathcal{M}^* = arg\,min_{\mathcal{M}_k}(\mathcal{M}_k\,)$$

(18)

The model with the lowest BIC was considered the most suitable in terms of both model complexity and ability to capture the underlying structure of the data [88].

The Elbow method and BIC are incorporated as baseline techniques for cluster estimation for classsical K-means, K-medoids, and GMM techniques to facilitate comparative evaluation.While commonly used, the Elbow Method, especially when relying on first-order derivatives ($\Delta$) often lacks robustness and precision. This highlights the need for a more adaptive approach. The proposed GCO-based optimization addresses this by effectively exploring the parameter space without relying on heuristic or visual methods.

## 2.7 Implementation configrarion

In this study, the proposed method is implemented in Python. While the K-means [112], K-medoids [113] and GMM [59] based clustering algorithms were implemented using python scikit-learn package, a widely adopted library for machine learning in Python. The computational evaluation were conducted on a Laptop equipped with an Intel Core i7 10th Generation processor, 16 GB of RAM, Windows 10 operating system and a solid-state drive (SSD) connected via the Serial Advanced Technology Attachment (SATA) interface.

**2.7.1 Evaluation measure.** To evaluate the performance of the clustering results, three evaluation measures are commonly used in existing literature including silhouette score, adjusted rand index and purity score. These evaluation metrics help in assessing the effectiveness of the unsupervised techniques and provide quantitative measures of its performance. Hence, we also incorporated these measures to evaluate the performance of our method.

### 2.7.1.1 Silhouette Score

The silhouette score offers an intuitive way to evaluate the quality of a clustering result by determining the mean silhouette coefficient across all samples [118,119]. This quantifies how well each data point fits within its assigned cluster compared to other clusters. [120]. Its value ranges from −1–1, where a higher silhouette score suggests better-defined, well-separated clustering results [121]. The silhouette score for a data point $i$ is defined as: a metric that quantifies how well each data point fits within its assigned cluster compared to other clusters.

$$S(i) = \frac{b(i) - a(i)}{\max(a(i), b(i))}$$

(19)

Where $a(i)$ represents the average distance between the point $i$ and all other points within the same cluster while $b(i)$ is the minimum average distance between the point $i$ and all points in the next nearest cluster [122]. However, the overall silhouette score for the entire dataset is calculated by the average of the silhouette scores for all the individual points.

### 2.7.1.2 Adjusted Rand Index (ARI)

The Adjusted Rand Index (ARI) is a widely used measure for evaluating the agreement between two data partitions in cluster analysis. It is often used to assess the quality of clustering algorithms [123–126], particularly between the obtained clustering results and ground-truth partition, ranging from 0 (complete disagreement) to 1 (complete agreement). Unlike the classical Rand Index, ARI adjusts the chance grouping of elements, making it a more robust and reliable measure of evaluating clustering results [127]. Mathematically it can be defined as:

$$ARI = \frac{RI - E[RI]}{\max(RI) - E[RI]}$$

(20)

Where $RI$ denotes the Rand Index that measures the agreement between two clustering results by counting pairs of data points that are either grouped together or assigned to different clusters in both the predicted and true clustering. Where $E[RI]$ represents the expected Rand Index, accounting for random chance.

The ARI has been generalized to compare a set of clustering solutions with a consensus matrix (ARImp) and to evaluate the consistency between two similarity matrices (ARImm), preserving desirable properties of the original ARI and introducing new ones, such as the ability to detect negative correlation and to compute in a distributed environment.

### 2.7.1.3 Purity Score

It is also a widely used external validation measure in clustering analysis, it measures the degree to which each cluster contains data points from a single ground-truth class. It represents how "pure" or homogeneous the clusters are in terms of the true labels, provide a simple technique to evaluate clustering quality. The Purity Score is defined as:

$$Purity\ Score = \frac{1}{n} \sum_{i=1}^{k} \max(|C_i \cap L_j|)$$

(21)

Where $C_i$ represents the set of points in ith cluster, and $L_j$ denotes the set of points in the ground-truth class $j$. Whereas the $\max(|C_i \cap L_j|)$ represents the largest number of points in cluster $i$ belonging to a single ground-truth class $L_j$.

### 2.7.1.4 F-measure

We also included the F-measure in our evaluation to provide a more reliable assessment of the proposed techniquue's performance. This can be mathematically defined as follows:

$$F - Measure = \frac{2}{\frac{1}{Precision} + \frac{1}{Recall}}$$

(22)

The computational time complexity of hybrid GM to identifying the clustering results is $O(t * K_i n_i D^2)$. Where t is the number of iterations, n is the number of data points, K is the number of components, and D is the dimensionality of the data. It can be observed that the dimensionality dominates the computation particularly for high-dimensional data sets. In E-step, computing the posterior probability $\gamma(\mathcal{Z}_{nk})$ requires $O(K_i n_i D)$ operations. In M-step, computing the parameters $\pi_k$, $\mu_k$ and $\Sigma_k$ involve $O(K_i n_i)$ operations. The estimation of D-dimensional mean vector of component k and $\Sigma_k$ variance matrix of component k required additional operations. Therefore, the overall time complexity is a combination of the complexities of both the Expectation-Maximization (EM) algorithm for Gaussian Mixture Models (GMM) and the Group Counseling Optimization (GCO) algorithm.

## 3. Results and discussion

In this research, our primary objective is to identify fake news on social media platforms based on unsupervised method, specifically utilizing a Gaussian Mixture Model (GMM). While it is well-known that the Expectation-Maximization (EM) algorithm, commonly used for fitting GMMs, is highly sensitive to initial values and requires the number of components to be specified a priori [128]. To address these challenges, we introduce a novel hybrid Gaussian Mixture model for detecting fake news in a fully unsupervised manner. Unlike previous works that separately address the issues of initialization [129] and the estimation of the number of components [130,131] we tackle these problems simultaneously.

The proposed approach also introduced a novel objective function for the GMM based on mixture distributions, and then leveraged a meta-heuristic optimization algorithm, Group Counseling Optimization (GCO) to find the best set of parameters to fit the Gaussian Mixture Model on news data. It iteratively adjusts the GMM parameters, ensuring that the model is both robust to initial values and capable of automatically determining the optimal number of clusters. This method significantly enhances the performance and reliability of the EM algorithm, thus, addressing the limitations of traditional GMM particularly sensitivity toward initial parameters values and number of cluster estimations, leading to more stable and optimal clustering. Finally, we also demonstrated the effectiveness of our innovative approach by applying it on a real-world news dataset.

In the initial step of the proposed Hybrid GMM, we initialized all the parameters of bounds vector as follows: the number of components (n_components) was adjusted within a range of 2–10, allowing the algorithm to identify the optimal number of clusters. This range was selected because a minimum of two components is necessary for meaningful clustering, and ten components represent a practical upper limit for this dataset. As there is a trade-off in the selection of an appropriate number of clusters. A higher number of clusters can offer a more detailed and fine-grained representation of the data, potentially enhancing clustering accuracy. However, a large number of clusters also leads to greater computational complexity and adds to the workload of post-processing [132]. The regularization on covariance (reg_covar) was adjusted between $10^{-6}$ and $10^{-2}$ ensuring that the covariance matrices remained stable and invertible while preventing numerical issues. This range provided a balance between too little and too much regularization, both of which could negatively impact the GMM's performance.

Furthermore, the maximum number of iterations (max_iter) was set between 100 and 500, providing the Expectation-Maximization (EM) algorithm with ample opportunity to converge while minimizing unnecessary computational overhead. The tolerance parameter (tol), ranging from $10^{-4}$ and $10^{-2}$, defined the convergence threshold for the EM algorithm, balancing the need for precision with the desire for computational efficiency. These carefully selected bounds allowed the GCO algorithm to effectively search for the optimal GMM parameters, leading to enhanced clustering performance and demonstrating the robustness of the approach in finding an optimal solution.

Following the optimization process, the GCO algorithm identified the optimal parameters for hybrid GMM, resulting in a model comprising of 3 components, with a regularization on covariance of 0.00617, a maximum of 224 iterations, and a tolerance of 0.00632. These optimized parameters were subsequently utilized to fit the hybrid GMM, superseding the initial values set within the bounds vector. Subsequently, the proposed hybrid GMM, configured with the identified 3 mixture

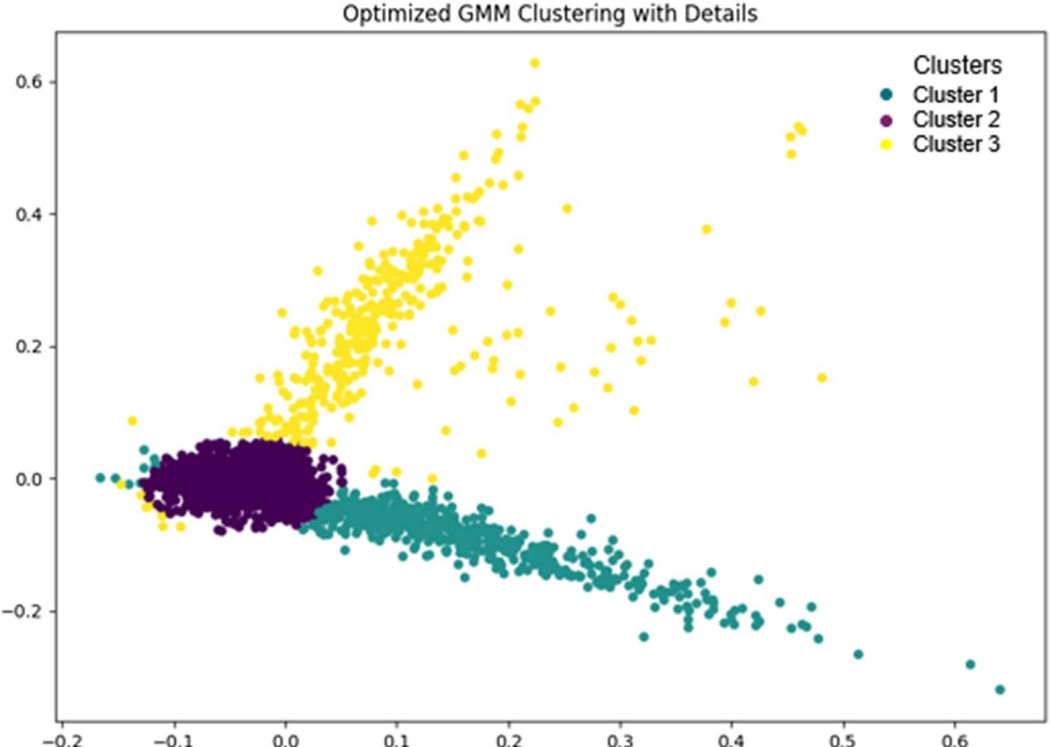

**Fig 4. Clustering results obtained by hybrid GMM over scaled data, showing distinct groupings of real and fake news articles based on textual similarity.**

components and the aforementioned optimized parameters is employed to cluster the two-dimensional unlabeled data, where each observation corresponds to a news statement. Upon fitting the hybrid GMM, the observations belonging to the same mixture component were considered as in the same cluster, facilitating the identification of underlying patterns in the news data. Fig 4 illustrated the clustering results produced by hybrid GMM over scaled data obtained from the preprocessing steps. It demonstrated that the proposed hybrid GMM partitioned the news data into specified number of clusters based on the mixture components as identified by the GCO optimizer and discussed above. A Superficial comparative analysis of Figs 2 and 4 might demonstrated that hybrid GMM primarily provides cluster delineation to the PCA representation, however, the integration of GMM with PCA offers substantial advantages that extend beyond mere visualization. It enables the modeling of data as a mixture of multiple Gaussian distributions, facilitating the identification of underlying data structures and probabilistic cluster assignments.

Nevertheless, the clustering results yielded by the hybrid Gaussian Mixture Model (GMM) are unlabeled meaning that while the data points are grouped into distinct clusters based on their features, the specific nature or identity of these clusters is not immediately discernible [64], as depicted in Fig 4. Thus, it does not provide any explicit indication of which cluster corresponds to which type of news. Furthermore, without labels, we are unable to ascertain the characteristics or thematic content of the clusters, nor can we directly identify the nature of the news articles within each cluster. Labeling these clusters would be a crucial step toward understanding the underlying structure of the data [133], as it would allow us to interpret the clusters in a meaningful way, such as identifying whether a cluster predominantly contains real or fake news. Thus, the process of labeling or annotating these clusters is essential for translating the unsupervised clustering results into actionable insights and for making informed decisions based on the classification of the news articles.

To address this challenge and to enhance the interpretability of the clusters generated by the hybrid GMM, a method leveraging the majority heuristic to label the clusters based on the ground truth was employed. This approach is pivotal in aligning the initially unlabeled clusters with the predefined categories within the dataset, thereby ensuring that the clustering outcomes are not only meaningful but also actionable.

Thus, the analysis of the ground truth labels associated with each data point, uniquely identified by an individual identifier, is conducted within each cluster. This process involves systematically evaluating the distribution of ground truth labels among the data points in each cluster. By determining the most prevalent or dominant ground truth label within a cluster, that label is designated as the representative identity for the entire cluster. For example, if the majority of data points within a cluster were labeled as "True," this label is assigned to the entire cluster, thus defining its identity.

The supplementary file presents a subset of data points (news) randomly sampled from each cluster, providing a representative overview of the clustering results. According to supplementary file, cluster 1, as demonstrated in the given Table, predominantly includes news statements labeled as "Mostly True". In contrast, the cluster 2 is characterized by a majority of news items labeled as either "False" or "Barely True." Furthermore, the cluster 3 is predominantly composed of news items labeled as "Pants on Fire," a designation reserved for statements that are not only false but are also egregiously incorrect or exaggerated. The consistent presence of these labels within a particular cluster highlights the effectiveness of the hybrid GMM algorithm in grouping news items that share a similar information.

A more thorough examination of the clustering results revealed that the proposed hybrid Gaussian Mixture Model (GMM) effectively consolidated the six subcategories of the news dataset into three primary categories. As previously discussed, Cluster 1 (green colored) primarily comprises news items labeled as "Mostly True," this cluster can be labeled as "Mostly Accurate" cluster. On the other hand, Cluster 2 (purple colored) is characterized by a majority of news items labeled as either "False" or "Barely True." Consequently, this cluster can be categorized under the "Partially Accurate" label, reflecting the presence of news items that, while containing some elements of truth, are either misleading or significantly distorted. Additionally, Cluster 3 (yellow colored), is predominantly composed of news items labeled as "Pants on Fire". As a result, this cluster can be labeled as "Inaccurate". Furthermore, it also simplifies the data, facilitating clearer distinctions between authentic and misleading information, which is crucial for the detection and analysis of fake information.

These categorizations illustrate the capability of the clustering algorithm to effectively distinguish among varying levels of truthfulness or misinformation in the news items and accurately reflecting the underlying structure of the dataset. Furthermore, by aligning clusters with these major categories, the analysis reinforces the robustness and reliability of the clustering approach, providing a clear understanding of the distribution of authentic and misleading information across the dataset. This structured categorization also enhances the potential for practical applications, such as misinformation detection and content validation, by enabling targeted analysis and intervention based on the identified cluster identities.

We also conducted a comprehensive set of evaluation to compare the performance of our proposed approach against traditional clustering algorithms, including Gaussian Mixture Model (GMM), K-means and K-medoids using the same news dataset. This comparative analysis was essential to evaluate the effectiveness and robustness of our method in contrast to classical clustering techniques.

During the evaluation each algorithm was meticulously applied under standardized conditions. The Bayesian Information Criterion (BIC) was utilized to ascertain the optimal number of components for traditional GMM [134]. For K-means and K-medoids clustering techniques, the Elbow method was employed to determine the optimal number of clusters, providing a visual assessment of the elbow point where adding more clusters fails to significantly improve the model [135].

The elbow point serves as a visual indicator for selecting the optimal number of clusters by analyzing the Within-Cluster Sum of Squares (WCSS) [133,134]. When applied to the news dataset, the Elbow method identified three clusters for K-means and five clusters for K-medoids, as shown in Fig 5. For K-means, it can be observed that increasing the number of clusters up to three as depicted in Fig 5(A) resulted in a significant reduction in WCSS (Elbow point), beyond which

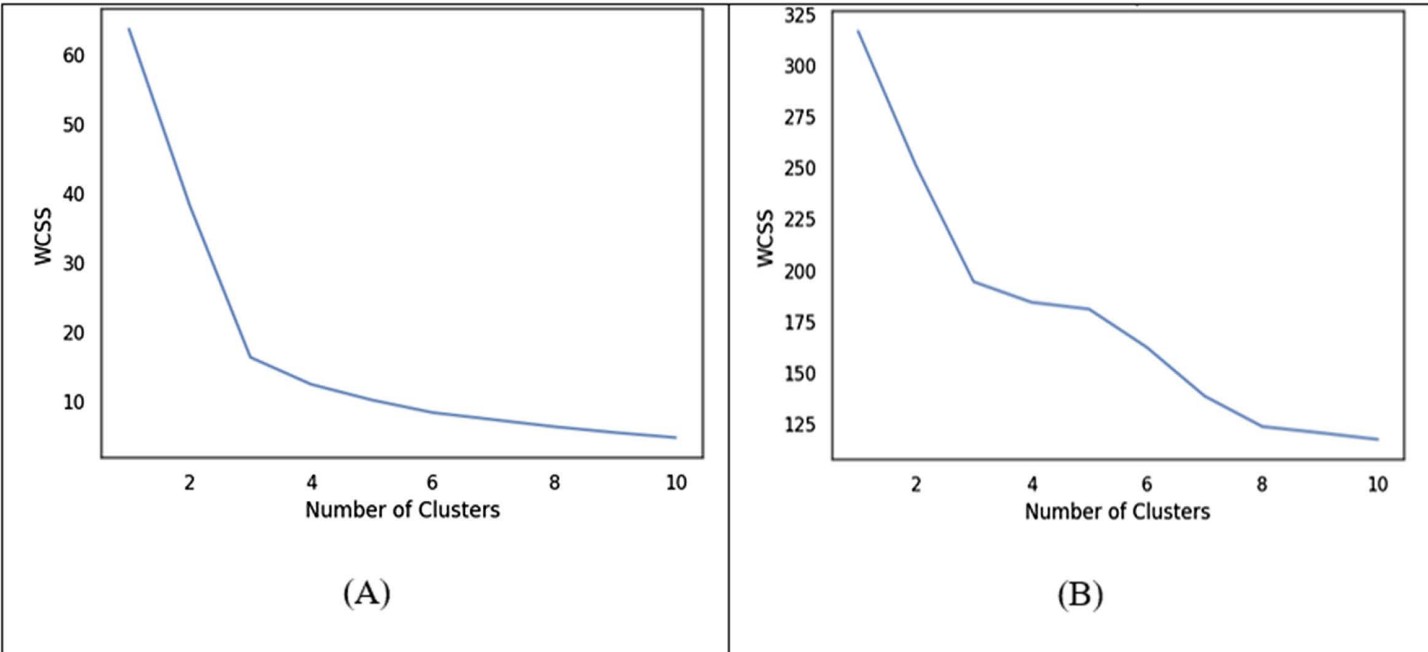

**Fig 5. Optimal Number of Cluster selection using Elbow method whereas (A) represent the Elbow graph for K-means to assess the Elbow point and (B) represent the Elbow graph for K-medoids clustering techniques respectively.**

the improvement becomes marginal. At this point, the trade-off between reducing WCSS and avoiding the unnecessary complexity of adding more clusters is balanced, ensuring that the selected model is efficient and well-fitted to the data and avoiding over-fitting. Similarly, for K-medoids, adding up to five clusters led to a substantial decrease in WCSS, after which further increases did not lead to noticeable improvement as depicted in Fig 5(B). Whereas BIC also suggested five clusters (components) for the classical GMM as can be seen in Fig 6. It also demonstrated that lower BIC scores indicating better model fit. The asterisk (*) highlights the configuration with the lowest BIC score, representing the optimal number of clusters and covariance structure for the GMM on the given news dataset.

This divergence in the optimal number of clusters across the algorithms highlights the sensitivity of each clustering technique to the underlying data. The justification related to the selection of these parameter selection techniques is detailed in the Methodology section.

Furthermore in the context of fake news detection, Figs 7 and 8 illustrates the clustering results obtained through traditional GMM and K-medoid clustering respectively, on the same dataset. These visual representations suggest a notable contrast in the distribution of data points between fake and true labels. Both traditional GMM and K-medoid identified five clusters, closely aligning with the five subcategories present in the news dataset. On the other hand, K-means discovered only three clusters as depicted in Fig 9. The evaluation results demonstrated that K-means algorithm tends to generalize the data, potentially merging subcategories that share similar characteristics, unlike GMM, which assigns data points based on probability distributions.

Furthermore, K-means algorithm minimize intra-cluster variance and effectively consolidates subcategories with shared characteristics into unified clusters. This integration is particularly beneficial as it reduces the complexity of the dataset and highlights broader patterns across related subcategories that exhibit surface-level similarities. It also provide a more comprehensive analysis of general trends and themes within the data. Such clustering can provide valuable insights into

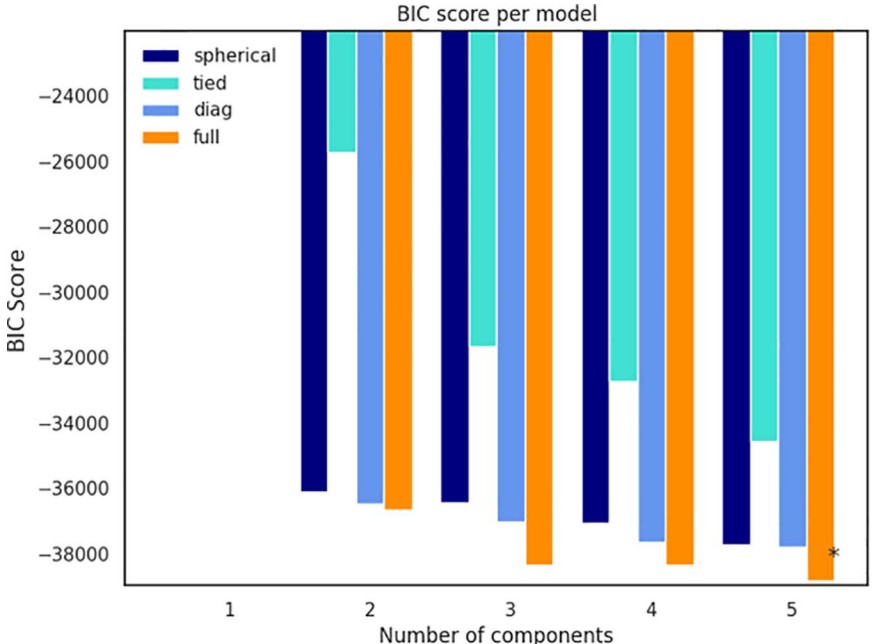

**Fig 6. Optimal number of cluster selection using BIC method for classical GMM clustering technique.**

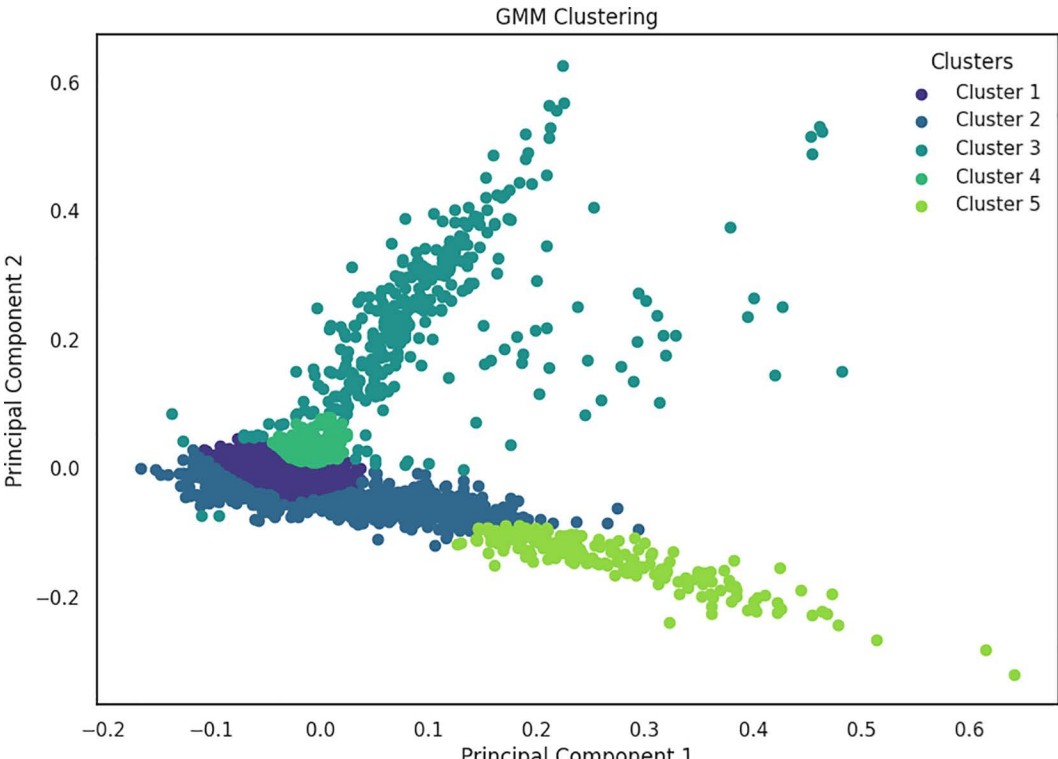

**Fig 7. Clustering results obtained using Classical GMM on scaled TF-IDF data, representing the distribution of news articles among different clusters based on textual similarities.**

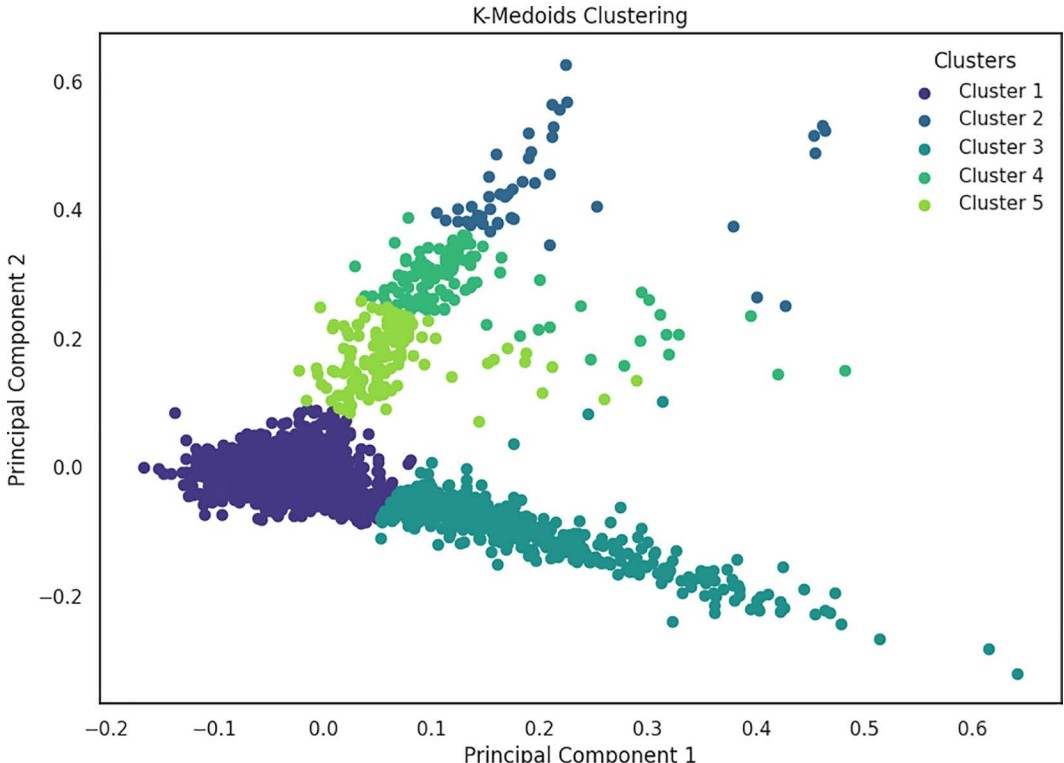

**Fig 8. Clustering results obtained using K-medoid.**

the structural relationships between different types of news content, enhancing the overall understanding of the dataset without compromising interpretability.

Therefore, to provide insight into the internal structure of clusters and their alignment with ground truth, we employed three performance measures including the Silhouette Score, Adjusted Rand Index (ARI), and Purity Score. Furthermore, the performance of these varied techniques in selecting the most suitable cluster count can be assessed through their respective Silhouette Score, Adjusted Rand Index (ARI), and Purity Score. Table 1 demonstrated a comparative analysis of the performance for each clustering algorithm.

The evaluation results demonstrated the superiority of our proposed Hybrid GMM, achieving a silhouette score of 0.77, which demonstrated more compact and well-separated clustering results, leading to better-defined cluster boundaries. The K-means algorithm also performed relatively well, with a silhouette score of 0.65, indicating that while its clusters are reasonably distinct, they are less cohesive than those produced by the Hybrid GMM.

The Classical GMM yielded a silhouette score of 0.61, showing moderate cluste r separation and cohesion. Although it performed better than K-medoids, its lower score compared to the Hybrid GMM reflects its inability to define well-separated clusters.

On the other hand, the K-medoid algorithm demonstrated the lowest performance, with a silhouette score of 0.52. This low score suggests that the clusters produced by K-medoid are less distinct and more prone to overlap, potentially leading to ambiguous clustering assignments. It can also be inferred that K-medoid may not be as effective in capturing the complex structure of the data. Furthermore, comparative analysis of evaluation results also depicted that hyprid GMM depicted an adjusted rand score of 0.83 and a purity score of 0.88. The purity score of hybrid GMM demonstrated a strong ability to form clusters with or homogeneous data points. On the other hand, K-means clustering algorithm yields 0.72 and 0.79 an

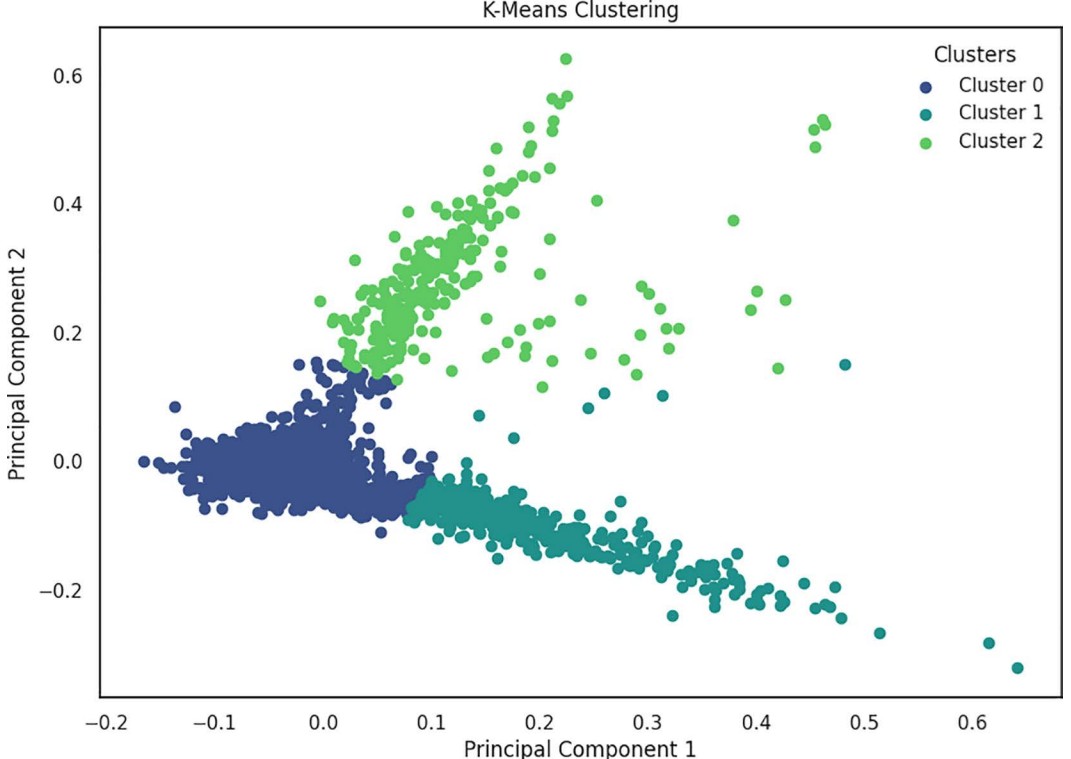

**Fig 9. Clustering results obtained using K-means.**

**Table 1. Comparative Analysis of evaluation metrics for Hybrid GMM, K-medoid, Classical GMM and K-means.**

| Algorithm | Silhouette Score | Adjusted Rand Index (ARI) | Purity Score | F-measure |
|---|---|---|---|---|
| **Hybrid GMM** | **0.77** | **0.83** | **0.88** | **0.85** |
| **K-medoid** | 0.52 | 0.62 | 0.73 | 0.69 |
| **Classical GMM** | 0.61 | 0.68 | 0.75 | 0.72 |
| **K-means** | 0.65 | 0.72 | 0.79 | 0.76 |

adjusted rand score and purity score respectively. Classical GMM and K- medoid clustering slightly performed poorly with an adjusted rand score and purity score of 0.68, of 0.75, 0.62 and 0.73 respectively.

In the context of clustering Fake News, the emphasis is often on achieving high Purity to ensure that most articles within a cluster are from the same category. Hybrid GMM, with the superior silhouette, adjusted rand score and purity appears to be the more effective model for this specific task. The present findings highlight substantial differences in clustering performance as measured by ARI on news data among the clustering techniques incorporated in this research. Therefore, based on the evaluation measures and the specific focus on clustering fake news, the proposed hybrid GMM emerges as significantly promising model for this analysis. Its promising performance across all the performance measures particularly purity score highlighted the Hybrid GMM's ability to effectively handle complex data, providing more reliable and well-defined clustering outcomes compared to classical GMM, k-means and k-medoid cluster methods incorporated in this research.

Furthermore, the proposed method is also aligned with real-world practical applications, especially in the realm of social media monitoring and content regulation. While the results are encouraging, the proposed technique has several

limitation. such as it does not incorporate community detection algorithms, which could aid in addressing the spread of misinformation and enable network-wide countermeasures. Moreover, the inclusion of hybrid models utilizing deep learning techniques and transformer-based architectures could improve detection precision by capturing intricate contextual subtleties within the text.

## Conclusion

This research proposed a hybrid Gaussian Mixture Model (GMM) for fake news detection. By leveraging Group Counseling Optimization (GCO), a meta-heuristic optimization algorithm, the proposed approach ensured robust parameter optimization for hybrid GMM and automatically determining the optimal number of clusters, significantly enhancing the reliability and performance of the Expectation-Maximization (EM) algorithm. The evaluation results also demonstrated that the proposed hybrid GMM outperformed traditional clustering methods, including classical GMM, K-means, and K-medoids. It can be concluded that the Hybrid GMM is a robust and promising solution for fake news detection, with practical potential for combating misinformation on social media platforms.

## Author contributions

**Conceptualization:** Sajida Perveen, Rehan Ashraf.

**Data curation:** Sami S. Albouq, Fatmah Alanazi, Rehan Ashraf.

**Formal analysis:** M. Usman Ashraf.

**Funding acquisition:** M. Usman Ashraf, Sami S. Albouq.

**Investigation:** Hanan E. Alhazmi, Fatmah Alanazi, Rehan Ashraf.

**Methodology:** Sajida Perveen.

**Project administration:** Khlood Shinan, Hanan E. Alhazmi, Muhammad Shahbaz.

**Resources:** Sami S. Albouq, Fatmah Alanazi.

**Software:** Sami S. Albouq.

**Supervision:** Muhammad Shahbaz.

**Validation:** Khlood Shinan, Muhammad Shahbaz.

**Visualization:** Khlood Shinan, Hanan E. Alhazmi.

**Writing – original draft:** Sajida Perveen.

**Writing – review & editing:** Khlood Shinan, Hanan E. Alhazmi, Muhammad Shahbaz.

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
