## [Decision Letter · Decision Letter 0]

22 Nov 2024

Dear Dr. Ashraf,

Thank you for submitting your manuscript to PLOS ONE. After careful consideration, we feel that it has merit but does not fully meet PLOS ONE’s publication criteria as it currently stands. Therefore, we invite you to submit a revised version of the manuscript that addresses the points raised during the review process.

We look forward to receiving your revised manuscript.

Kind regards,

Fredrick Romanus Ishengoma

Academic Editor

PLOS ONE

Journal Requirements:

Reviewers' comments:

Reviewer's Responses to Questions

**Comments to the Author**

1. Is the manuscript technically sound, and do the data support the conclusions?

Reviewer #1: Partly

Reviewer #2: Partly

2. Has the statistical analysis been performed appropriately and rigorously?

Reviewer #1: No

Reviewer #2: No

3. Have the authors made all data underlying the findings in their manuscript fully available?

Reviewer #1: No

Reviewer #2: No

4. Is the manuscript presented in an intelligible fashion and written in standard English?

Reviewer #1: Yes

Reviewer #2: Yes

Reviewer #1: This article presents work on unsupervised fake news detection.

I find that the work presented in this manuscript is not complete, the authors barely scratched the surface, and this work falls short of being ready for publication.

The methods used are not novel at all.

1. I find that the authors have ignored a lot of related literature on the subject that proposes new ways of feature extraction.

Please read, present, and discuss at least related work regarding the following aspects (please respond individually):

- the use effect of word embeddings to encode fake news [1]

- use of transformer embeddings to encode fake news [2]

- how document embeddings are used to encode fake news [3]

- real-time architectures for fake news and harmful content detection and mitigation [4] (e.g., ContCommRTD https://doi.org/10.1109/TKDE.2024.3417232) or [5] (e.g. StopHC https://arxiv.org/abs/2411.06138(

- using social network features for fake news detection [6] (e.g., https://doi.org/10.1016/j.knosys.2024.111715)

- the effect of multilingual transformers on fake news detection [7] (e.g., http://ceur-ws.org/Vol-3180/paper-61.pdf)

2. What happens after detection? There are many solutions used for this proposed in the literature that also propose models for identifying fake news and mitigation.

Some related work that should be discussed (please respond individually):

- use of community detection for fake news network immunization [8] (e.g., https://doi.org/10.1016/j.jestch.2024.101728)

- use of weighted directed spanning trees for fake news detection and mitigation in real time [9] (e.g., https://doi.org/10.1109/ACCESS.2023.3331220)

- use of budget-based immunization algorithms to stop fake news from spreading [10] (e.g., https://doi.org/10.1145/3459637.3482481)

3. There are many methods for document similarity detection, not just clustering. The manuscript must be improved and it must discuss the following (please respond individually):

a. Document similarity using clustering for dimensionality reduction [11] (e.g., https://doi.org/10.1109/AQTR49680.2020.9129967)

b. Novel novel density-based clustering algorithms that can be used for topic modeling that should at least be mentioned, e.g., DenLAC [12], some even based on DBScan algorithms [13], for example, QuickDBScan and KDTreeDBSCAN [14].

c. Community Detection-based methods for document similarity [15].

d. Similarity based on context-aware document clustering [16].

e. How are the metrics for encoding the textual data selected? Please compare with other methods, you can start from [17] (e.g., http://doi.org/10.1109/SYNASC.2016.055)

4. An in-depth exploratory data analysis is required.

5. The experiments should be improved (please respond individually to each point in the answer to reviewers)

a. There is no hyperparameter tuning for the proposed model.

b. There is no time performance evaluation.

c. The experiments are done on relatively small datasets. How does the proposed model perform on a larger dataset?

d. There is no comparison with state-of-the-art models.

6. The conclusions are very shallow.

7. What are the limitations of this study?

8. For reproducibility purposes, the authors should make the code publicly available. Without publicly available source code, models, and datasets, a shadow of doubt can fall on the results of this work. How can anyone check if these results are real or made up without the code, models, and datasets?

9. Please proofread and spell-check the article before resubmitting.

[1] https://scholar.google.com/scholar?q=misinformation+detection+word+embeddings

[2] https://scholar.google.com/scholar?q=transformers+misinformation

[3] https://scholar.google.com/scholar?q=fake+news+document+embeddings

[4] https://scholar.google.com/scholar?q=content-based+misinformation+real-time

[5] https://scholar.google.com/scholar?q=content+detection+mitigation+architecture+social+media

[6] https://scholar.google.com/scholar?q=deep+neural+network+ensemble+social+context+fake+news+detection

[7] https://scholar.google.com/scholar?q=fake+news+detection+sentence+transformer

[8] https://scholar.google.com/scholar?q=community+algorithm+network+immunization+fake+news

[9] https://scholar.google.com/scholar?q=tree+algorithm+real-time+fake+news+mitigation+social+media

[10] https://scholar.google.com/scholar?q=social+network+immunization+harmful+speech

[11] https://scholar.google.com/scholar?q=document+vector+model+clustering+dimensionality+reduction

[12] https://scholar.google.com/scholar?q=DenLAC

[13] https://scholar.google.com/scholar?q=density+text+clustering+document+embeddings

[14] https://scholar.google.com/scholar?q=evaluation+DBSCAN+similarity+join

[15] https://scholar.google.com/scholar?q=community+detection+document+clustering

[16] https://scholar.google.com/scholar?q=topic+modeling+contextual+cues

[17] https://scholar.google.com/scholar?q=comparing+term+weighting+topic+modeling

Reviewer #2: The authors propose to detect fake news using an unsupervised detection approach.

In its current state, the manuscript is not ready for publication and it requires improvements.

The proposed method combines existing methods.

1. Why choose TF-IDF for encoding the documents?

Why not word embeddings [1], transformers [2], or document embeddings [3]?

2. Does the method work in real-time? Please compare with existing methods and architectures that work in real-time [4] or [5].

3. What happens when encoding more than the textual content? For example, if using social network features for fake news detection [6]

4. How do you deal with multilingual textual data? Compared with existing methods [7].

5. What's next after detection? In the current literature, many works propose models for fake news detection and mitigation, e.g., community-based [8], trees-based [9], or pre-emptive [10] algorithms. The authors must discuss some direction at least in future work.

6. Why use only classic clustering methods for your approach? There are many clustering methods that employ dimensionality reduction [11]. Please improve the manuscript and compare your methods with these methods.

7. On the topic of clustering, why not use density-based algorithms [12] (e.g., DenLAC) or DBScan algorithms [13], for example, QuickDBScan and KDTreeDBSCAN [14]. Please improve the manuscript and compare your methods with these methods.

8. Community Detection-based methods for document similarity [15] are also used for clustering documents. Please improve the manuscript and compare your methods with these methods.

9. Why not use topic modeling for clustering documents? There are many context-aware document topic modeling algorithms [16] used in the literature for document clustering. Please improve the manuscript and compare your methods with these methods.

10. On what basis was TF-IDF selected for encoding the textual data? There are many works that discuss different weighting schemes for document clustering [17]. Please improve the manuscript and compare your methods with these methods.

11. The datasets are not properly presented. Please improve the manuscript with an exploratory data analysis.

12. The experiments are missing a hyperparameter tuning analysis.

13. The experiments are missing a time performance evaluation.

14. The size of the datasets is limited, how does the proposed model perform on a large dataset?

15. Improve the manuscript by comparing your method with existing state-of-the-art models.

16. The conclusions must be improved.

17. A limitation section is missing. What are the limitations of your methods?

18. The authors should make the code publicly available for reproducibility purposes.

19. Before resubmitting, I recommend proofreading and spell-checking the article.

[1] https://scholar.google.com/scholar?q=misinformation+detection+word+embeddings

[2] https://scholar.google.com/scholar?q=transformers+misinformation

[3] https://scholar.google.com/scholar?q=fake+news+document+embeddings

[4] https://scholar.google.com/scholar?q=content-based+misinformation+real-time

[5] https://scholar.google.com/scholar?q=content+detection+mitigation+architecture+social+media

[6] https://scholar.google.com/scholar?q=deep+neural+network+ensemble+social+context+fake+news+detection

[7] https://scholar.google.com/scholar?q=fake+news+detection+sentence+transformer

[8] https://scholar.google.com/scholar?q=community+algorithm+network+immunization+fake+news

[9] https://scholar.google.com/scholar?q=tree+algorithm+real-time+fake+news+mitigation+social+media

[10] https://scholar.google.com/scholar?q=social+network+immunization+harmful+speech

[11] https://scholar.google.com/scholar?q=document+vector+model+clustering+dimensionality+reduction

[12] https://scholar.google.com/scholar?q=DenLAC

[13] https://scholar.google.com/scholar?q=density+text+clustering+document+embeddings

[14] https://scholar.google.com/scholar?q=evaluation+DBSCAN+similarity+join

[15] https://scholar.google.com/scholar?q=community+detection+document+clustering

[16] https://scholar.google.com/scholar?q=topic+modeling+contextual+cues

[17] https://scholar.google.com/scholar?q=comparing+term+weighting+topic+modeling

**Do you want your identity to be public for this peer review?** For information about this choice, including consent withdrawal, please see our Privacy Policy

Reviewer #1: No

Reviewer #2: No

---

## [Author Response · Author response to Decision Letter 1]

5 Dec 2024

Original Manuscript ID: PONE-D-24-50144

Original Article Title: “Unsupervised fake news detection on social media using hybrid gaussian mixture model”

To: PLOS ONE Editor

Re: Response to reviewers

Dear Editor,

Thank you for allowing a resubmission of our manuscript, with an opportunity to address the reviewers’ comments.

We are uploading (a) our point-by-point response to the comments (below) (response to reviewers, under “Response-to-Reviewers Files”), (b) an updated manuscript with yellow highlighting indicating changes (as “Revised Article with Changes Highlighted”), and (c) a clean updated manuscript without highlights (“Manuscript”).

Best regards,

Reviewer 1, Concern # 1. (Please read, present, and discuss at least related work regarding the following aspects (please respond individually):- the use effect of word embeddings to encode fake news [1]- use of transformer embeddings to encode fake news [2]- how document embeddings are used to encode fake news [3]- real-time architectures for fake news and harmful content detection and mitigation [4] (e.g., ContCommRTD https://doi.org/10.1109/TKDE.2024.3417232) or [5] (e.g. StopHC https://arxiv.org/abs/2411.06138(- using social network features for fake news detection [6] (e.g., https://doi.org/10.1016/j.knosys.2024.111715)- the effect of multilingual transformers on fake news detection [7] (e.g., http://ceur-ws.org/Vol-3180/paper-61.pdf)):

Author response: We appreciate the insightful feedback from the reviewer specifically focusing on different type of embedding techniques and their role in our proposed methodology. The point-by-point response to the comments is given below:

The use effect of word embeddings to encode fake news

The use of word embeddings like Word2Vec and GloVe for fake news detection has been extensively studied due to their ability to capture semantic relationships between words. However, there are specific reasons why these embeddings were not included in the proposed research:

Word embeddings generate static word vectors, which fail to account for contextual nuances, such as changes in meaning based on surrounding text [1, 2]. While effective in supervised learning tasks, these are less suited for unsupervised methods like clustering, where the focus is on structural and statistical characteristics of the data. Approaches like TF-IDF provide term-specific weights that are more interpretable and better aligned with the objectives of the proposed research based on clustering [3]. Furthermore, word embeddings typically require additional preprocessing and fine-tuning to adapt them for domain-specific fake news detection. This process adds computational complexity and might not yield significant improvements for unsupervised learning methods [4]. The proposed research incorporated TF-IDF, being simpler and domain-agnostic, offers a scalable solution without the overhead of embedding model training or fine-tuning [5].

Use of transformer embeddings to encode fake news

Transformer embeddings are typically leveraged in supervised learning settings or fine-tuned for specific tasks. The proposed research incorporated unsupervised clustering techniques, where traditional representations like TF-IDF are often more effective and interpretable for capturing textual patterns [6]. Another key challenge associated with transformer-based models is their lack of transparency and interpretability. These models generate high-dimensional embeddings that can be difficult to analyze and interpret in the context of clustering, where understanding the relationship between data points is critical [7].

How document embeddings are used to encode fake news

Document embeddings are a powerful tool for encoding fake news by transforming textual data into numerical representations that capture the semantic and contextual essence of entire documents. Unlike traditional word-level embeddings, document embeddings aggregate information across sentences or paragraphs, making them particularly effective for tasks requiring an understanding of broader context.

However, the primary goal of this research is to analyze surface-level linguistic patterns for fake news detection. Therefore, traditional methods like TF-IDF were sufficient to capture the necessary textual features without the additional complexity of embedding-based approaches.

Real-time architectures for fake news and harmful content detection and mitigation

The proposed research primarily investigates unsupervised learning techniques, specifically GMM-GCO, a hybrid model to effectively identify fake news. Real-time architectures often involve supervised or semi-supervised methods, leveraging labeled data streams for real-time predictions. Integrating these techniques could deviate from the core aim of developing robust unsupervised methods for textual clustering. Furthermore, real-time feedback systems, such as those used in platforms like ContCommRTD or StopHC could demand infrastructure beyond the scope of the proposed research to implement and test these architectures

Using social network features for fake news detection

Real-time systems incorporate interdisciplinary components, such as user behavior analysis, network traffic monitoring, and large-scale content moderation, which is beyond the scope of our research. As the primary goal of this research is to analyze surface-level linguistic patterns for fake news detection.

The effect of multilingual transformers on fake news detection

Our study primarily focused on English-language datasets and align with several existing benchmark corpora in fake news detection studies. Since multilingual transformers like mBERT or XLM-R are designed to handle diverse languages, their inclusion is not necessary given the considered dataset's linguistic homogeneity. Multilingual models like mBERT and XLM-R have shown success in cross-lingual natural language understanding tasks, but their effectiveness in unsupervised or clustering techniques remains less explored [8, 9].

Author action:

We trust that these justifications align with the reviewer’s expectations and justify the exclusion of word or transformer embeddings in favor of simpler, domain-independent text representation techniques better aligned with the objectives of our proposed research.

References:

1. Mikolov T, Sutskever I, Chen K, Corrado GS, Dean J. Distributed representations of words and phrases and their compositionality. Advances in neural information processing systems. 2013;26.

2. Sun C, Yang Z, Luo L, Wang L, Zhang Y, Lin H, Wang J. A deep learning approach with deep contextualized word representations for chemical–protein interaction extraction from biomedical literature. IEEE Access. 2019 Oct 21;7:151034-46.

3. Sparck Jones K. A statistical interpretation of term specificity and its application in retrieval. Journal of documentation. 1972 Jan 1;28(1):11-21.

4. Farhangian F, Cruz RM, Cavalcanti GD. Fake news detection: Taxonomy and comparative study. Information Fusion. 2024 Mar 1;103:102140.

5. Pennington J, Socher R, Manning CD. Glove: Global vectors for word representation. InProceedings of the 2014 conference on empirical methods in natural language processing (EMNLP) 2014 Oct (pp. 1532-1543).

6. Park J, Park C, Kim J, Cho M, Park S. ADC: Advanced document clustering using contextualized representations. Expert Systems with Applications. 2019 Dec 15;137:157-66.

7. Li Y, Wang J, Dai X, Wang L, Yeh CC, Zheng Y, Zhang W, Ma KL. How does attention work in vision transformers? A visual analytics attempt. IEEE transactions on visualization and computer graphics. 2023 Mar 27;29(6):2888-900.

8. Conneau A, Baevski A, Collobert R, Mohamed A, Auli M. Unsupervised cross-lingual representation learning for speech recognition. arXiv preprint arXiv:2006.13979. 2020 Jun 24.

9. Pires T. How multilingual is multilingual BERT. arXiv preprint arXiv:1906.01502. 2019.

Reviewer#1, Concern # 2. (What happens after detection? There are many solutions used for this proposed in the literature that also propose models for identifying fake news and mitigation.

Some related work that should be discussed (please respond individually):

- use of community detection for fake news network immunization [8] (e.g., https://doi.org/10.1016/j.jestch.2024.101728)

- use of weighted directed spanning trees for fake news detection and mitigation in real time [9] (e.g., https://doi.org/10.1109/ACCESS.2023.3331220)

- use of budget-based immunization algorithms to stop fake news from spreading [10] (e.g., https://doi.org/10.1145/3459637.3482481).):

Author response: We sincerely appreciate the valuable feedback provided by the reviewer, a point by point response for each comment is given below:

Use of community detection for fake news network immunization

The proposed research primarily focuses on unsupervised clustering techniques and hybrid GMM methods for detecting fake news based on linguistic patterns in textual data. Community detection in fake news network immunization is more relevant to network-based methods, where relationships between social media accounts or information dissemination pathways are analyzed. This network-based focus falls outside the linguistic and clustering scope of your study. Community detection approaches rely on graph-based representations of social networks, where nodes represent users or sources and edges represent interactions or information flows. In contrast, our study employs textual data and uses TF-IDF vectorization combined with Gaussian Mixture Models (GMM) and Group Counseling Optimization (GCO). The lack of graph-structured data in the proposed research makes the integration of community detection techniques unsuitable.

Community detection frameworks often propose mitigation strategies like node immunization to curb the spread of fake news by identifying and neutralizing influential nodes. While valuable, such approaches require a detailed understanding of network dynamics, user influence, and propagation pathways, which is out of the scope of our research.

Use of weighted directed spanning trees for fake news detection and mitigation in real time

Our research primarily emphasizes the detection of fake news using unsupervised clustering techniques such as GMM, K-means, K-medoid and GCO-GMM, focusing on linguistic patterns in textual data. The proposed method focused on transforming text into numerical features (e.g., TF-IDF) and applying clustering algorithms to identify patterns. In contrast, approaches using weighted directed spanning tree (WDST) often operate on graph-based representations of information spread, focusing on the propagation of fake news across networks, which is beyond the scope of the proposed research.

Furthermore, our dataset does not include interaction or network related information (such as user engagement data, timestamps, or propagation paths) required for weighted directed spanning trees methods. The absence of such data makes it impractical to integrate WDST based methodologies into in the proposed research.

Use of budget-based immunization algorithms to stop fake news from spreading

Our proposed research is fundamentally designed to address fake news detection through unsupervised clustering techniques, particularly focusing on linguistic and textual patterns in news data. Budget-based immunization algorithms, which are primarily used in network-based propagation control, fall outside the scope of the proposed research. These algorithms focus on identifying key nodes (users or sources) in a social network to block or mitigate the spread of misinformation, requiring a network representation and interaction data.

Furthermore, our dataset does not include interaction or network related information (such as user engagement data, timestamps, or propagation paths) required for weighted directed spanning trees methods. The absence of such data makes it impractical to integrate WDST based methodologies in the proposed research.

Author action: We sincerely hope that these clarifications will meet the reviewers' expectations and effectively address their concerns.

Reviewer#2, Concern # 3. (There are many methods for document similarity detection, not just clustering. The manuscript must be improved and it must discuss the following (please respond individually):

a. Document similarity using clustering for dimensionality reduction [11] (e.g., https://doi.org/10.1109/AQTR49680.2020.9129967)

b. Novel novel density-based clustering algorithms that can be used for topic modeling that should at least be mentioned, e.g., DenLAC [12], some even based on DBScan algorithms [13], for example, QuickDBScan and KDTreeDBSCAN [14].

c. Community Detection-based methods for document similarity [15].

d. Similarity based on context-aware document clustering [16].

e. How are the metrics for encoding the textual data selected? Please compare with other methods, you can start from [17] (e.g., http://doi.org/10.1109/SYNASC.2016.055)).

a. Document similarity using clustering for dimensionality reduction

The primary objective of the proposed research is to optimize GMM parameters using GCO for improved clustering results and efficiently detection of fake news. Clustering for dimensionality reduction, while valuable, does not directly contribute to the parameter optimization goals.

Furthermore the proposed research already employs TF-IDF and PCA for feature extraction and dimensionality reduction (Please see the highlighted content under 2. 1. 8 heading on page 10 and 11). These techniques have strong theoretical and empirical implications in fake news detection, and introducing clustering for dimensionality reduction might not add substantial value over the current methods.

The focus on fake news detection through clustering assumes a direct clustering algorithm application. Clustering for dimensionality reduction serves a more generalized purpose and could dilute the research scope.

b. Novel density-based clustering algorithms that can be used for topic modeling that should at least be mentioned, e.g., DenLAC [12], some even based on DBScan algorithms [13], for example, QuickDBScan and KDTreeDBSCAN [14].

Our prior experience with DBSCAN yielded unsatisfactory results in terms of clustering accuracy for fake news detection. Although these newer density-based algorithms (QuickDBScan, KDTreeDBSCAN and DenLAC) may offer improvements, the reason to exclude them was likely based on the failure of DBSCAN to meet the requirements for our specific dataset and the added complexity these algorithms could introduce without guaranteed improvements.

The primary focus of the proposed research is on optimizing the parameters of the Gaussian Mixture Model (GMM) using the Group Counseling Optimization (GCO) method. As GMM offers a more probabilistic approach to clustering, that has the potential to capture the uncertainty of fake news data more effectively, it was deemed more suitable for our goals than attempting to integrate another clustering method that may require additional optimization efforts.

c. Community Detection-based methods for document similarity

The proposed research prioritizes optimizing GMM for clustering due to its probabilistic nature. While community detection offers graph-based insights, it differs significantly in its underlying assumptions and might not align directly with our optimization framework.

Furthermore, our dataset does not include interaction or network related information (such as user engagement data, timestamps, or propagation paths) required for graph-based methods. The absence of such data makes it impractical to integrate graph based methodologies in the proposed research.

d. Similarity based on context-aware document clustering

The use of word embeddings like Word2Vec and GloVe for fake news detection has been extensively studied due to their ability to capture semantic relationships between words. However, there are specific reasons why these embeddings were not included in the proposed research:

Word embeddings generate static word vectors, which fail to account for contextual nuances, such as changes in meaning based on surrounding text [1, 2]. While effective in supervised learning tasks, word embeddings are less suite

---

## [Decision Letter · Decision Letter 1]

12 Mar 2025

Dear Dr. Ashraf,

Thank you for submitting your manuscript to PLOS ONE. After careful consideration, we feel that it has merit but does not fully meet PLOS ONE’s publication criteria as it currently stands. Therefore, we invite you to submit a revised version of the manuscript that addresses the points raised during the review process.

We look forward to receiving your revised manuscript.

Kind regards,

Fredrick Romanus Ishengoma

Academic Editor

PLOS ONE

Reviewers' comments:

Reviewer's Responses to Questions

**Comments to the Author**

Reviewer #3: All comments have been addressed

Reviewer #4: (No Response)

Reviewer #5: (No Response)

Reviewer #6: All comments have been addressed

Reviewer #7: (No Response)

2. Is the manuscript technically sound, and do the data support the conclusions?

Reviewer #3: Yes

Reviewer #4: Partly

Reviewer #5: Partly

Reviewer #6: Yes

Reviewer #7: Yes

3. Has the statistical analysis been performed appropriately and rigorously?

Reviewer #3: Yes

Reviewer #4: N/A

Reviewer #5: N/A

Reviewer #6: No

Reviewer #7: N/A

4. Have the authors made all data underlying the findings in their manuscript fully available?

Reviewer #3: Yes

Reviewer #4: No

Reviewer #5: Yes

Reviewer #6: Yes

Reviewer #7: Yes

5. Is the manuscript presented in an intelligible fashion and written in standard English?

Reviewer #3: Yes

Reviewer #4: No

Reviewer #5: Yes

Reviewer #6: Yes

Reviewer #7: Yes

Reviewer #3: The authors deal with an interesting topic, which could be of benefit to specific journal collection, the topic is well work out at the required level in term of content and of formal aspect:

- the aims are clear formulated;

- related work is worked out at the good level, the used of the resources are sufficient (138 resources);

- the paper is well written, it meets the formal requirements;

- the extent of the paper is adequate, the structure is clear and the conclusions are sufficient.

Reviewer #4: The paper is poorly prepared. Few fundamental writing issues are mentioned below.

1.

Abstract is filled by useless background text starting from the "advent of World Wide Web" and its revolutionary rise making me doubt which century this paper was written. There are also typos (reserch) indicating that nobody has put serious efforts for the writing. This is amazing since there are 8 authors in the paper. It takes just couple of sentences to define the research question (fake news detection), and the proposed approach (unsupervised clustering) and the novelty (measure to select the number of clustrers).

2.

Introduction focus on superficial literature review, and near the end, suddenly jumping from the backgroud to explaining that Gaussian mixture models has been widely used modeling dynamic systems without telling what is the dynamic system and how it relates to the previous text.

3.

Section 2 is non-innovatively titles as "Proposed methodology" covering everything from pre-processing (TF-IDF is hardly pre-processing anymore), to Gaussian mixture models and selecting the number of clusters including four levels of sections showing the complete failure to organize the writing properly. The results then follow in section which is not even numbered.

4.

GCO is claimed to be used to solve the hyperparameters like the number of components and converging threshold but later Elbow method and BIC are introduced anyway. Why? Elbow method seems to refer calculating first derivative (delta), which is not sufficient to alone for the task.

5.

The more essential are missing from the paper. Why (and how) we can expect to detect fake news from the clustering result, or Gaussian mixture models? None of the pictures show the true/fake labels for the points, only there cluster id.

6.

Figures are missing from the paper but included separately making it hard to follow. The document is also mixed by review reports, responses, two versions of the paper, in seemingly random order. Not sure if this is automated or author organized order of the material, but it just does not work well.

Overall, I cannot sure has any real research done. The text might be just written by AI, added some fake graphs and curves. At least I did not find any evidence of any intellectual human involvment (despite of 8 names). On the contrary, seems there is lack of human involvmenent in the process here. I am amazed this kind of paper is even allowed to be revised. Reject straightforward.

Reviewer #5: Referee report

Re: PONE-D-24-50144R1

In this work, clustering algorithms are applied and compared to provide an unsupervised method of detecting fake news in social media. Specifically, a combined method that was shown to be efficient in this field was proposed. The authors have submitted a revision of their original manuscript addressing the comments of the first referee round. However, though they have replied in detail (including respective references) directly to the comments in the rebuttal letter, few of these comments were addresses in the revised manuscript. At the same time, it does not seem to be any page 19 or 20 in the manuscript on which new content was added - see e.g. reply to Reviewer 1, Concern #5). The authors should consider addressing the comments in their manuscript and add there the new references. Especially, the comments on applicability, conclusions, reproducibility, and limitations of this study should be addressed in detail.

In that respect, the manuscript should be considered for publication in PLOS One after all comments of the previous and current referee round are carefully addressed (also in the manuscript).

Comments

1. Balancing the data: were class balancing algorithms used for this purpose or data were chosen based on a random number generator or similar? Please, clarify. If that is the case, why weren't class balancing algorithms used? Also, what parts of the data are used: are these vectors of words and/or TF-IDF values?

2. PCA analysis: please clarify the input data for this. What is the dimension of the post-processed data? Are only the words in form of a vector used for the PCA?

3. Proposed hybrid GMM model: is this taking as input only the 2D data from the PCA analysis? Please, clarify, what data are used exactly. On pg. 8 it is stated that the fitness is based on a comparison of predicted to true labels. However, the authors claim that their method is unsupervised. Please, clarify in detail the process. Also, the term 'agent' should be defined clearly.

4. Why were both the elbow method and the Silhouette score used? Both provide the number of clusters. One of these should be sufficient.

5. On pg.12 reference on the posterior probability, E-step and M-step is made without providing any details of these terms. These should be clarified in the context of this work.

6. Some benchmark numbers on the computational performance of the hybrid GMM model should be provided, i.e. resources used and computing time for the analysis.

7. Comparison of Fig.2 and 4 reveals that the only additional information obtained from the hybrid GMM is the division of the representation in clusters. Is the whole computational effort worth it or could the PCA analysis be used in another more efficient way to reach the detection goal? The authors should clarify this to strengthen their proposed model.

8. The clusters in Fig.4 were assigned a label based on the labels in the original data set. How does then the statement of unsupervision hold? Is the whole model applied only to go back to assign labels from the original set? That supports the fact that the labels are needed.

Also parts of the data of cluster 1 and 3 are quite spread and include many outliers. How can these be handled in terms of fake news detection?

9. Fig.5: the elbow method seems not to be efficient for the k-Medoid method, as there are two abrupt changes in the curve.

10. The authors state on pg.15 that "This divergence in the optimal number of clusters across the algorithms highlights the sensitivity of each clustering technique to the underlying data." This would be a very important factor that would hinder the use of the proposed method for the detection of fake news, as the detection would be prone to large errors based on the number of clusters detected. How can the authors overcome this view?

11. Comparing Fig. 4 and 9: it can be clearly seen that k-means performs the same as the hybrid GMM and should also be computationally faster. How does then the hybrid GMM method perform better, as implied in the manuscript on pg.16and in the conclusions?

12. Finally, it is not clear how the hybrid GMM model could realistically contribute in fake news detection. First, it does not provide additional valuable information beyond the k-means algorithm. At the same time, the detected clusters need to be labeled to distinguish between 'fake' and 'true' news. In that case, it seems to be practically of higher interest to apply an efficient supervised scheme that can directly classify the news, even if that is computationally more demanding. In that respect, the authors should either strengthen the arguments used and analysis performed or direct their proposed methods to other applications.

Minor Comments

(a) Some typesetting errors throughout should be corrected, e.g. 'vactorization' on pg.6.

(b) Provide references - were missing - for the algorithms used, e.g. GMM (pg.7) or Bayesian Information Criterion (BIC) or k-Medoids (pg.10), when these are first introduced. The same holds for other used concepts/codes/libraries, like scikit (pg.11).

(c) Define abbreviations when first used, e.g. PDF on pg.7.

(d) The title of subsection 2.7 should be changed, as there are no experiments involved.

(e) Pg. 15: "We also conducted a comprehensive set of experiments" and "During the experiments", please rephrase as no experiments were performed. The same on all pages referring to 'experiments'.

(f) The legends in Fig.6 are not clear, please define.

Reviewer #6: 1. The paper introduces a novel hybrid approach combining GMM with GCO for unsupervised fake news detection. This is addresses the limitations of traditional GMM, such as sensitivity to initial parameters and the need for manual cluster selection. However, the paper could benefit from a more detailed discussion on how this hybrid approach compares to other state-of-the-art unsupervised methods in fake news detection.

2. The methodology section should provide a more detailed explanation of the GCO algorithm and its integration with the GMM. Specifically, the paper should elaborate on how GCO avoids local optima through its population-based optimization approach and how it enhances GMM's performance in fake news detection by automating parameter tuning and improving convergence to globally optimal solutions.

3. The paper uses Silhouette Score, Adjusted Rand Index, and Purity Score for evaluation. However, the paper should also consider including other metrics like F1-score or precision/recall to provide a more comprehensive evaluation of the model's performance.

4. The paper should provide more details on the data preprocessing steps, such as how the dataset was balanced and why certain preprocessing techniques were chosen.

5. Ensure all figures, tables, and equations are referenced in the text. For example, Fig. 5 and Fig. 6 are not discussed in detail in the results section.

6. In Figure 4 and Figure 7 captions lack sufficient detail. For example, Fig. 4 should specify what the clusters represent.

7. In table 1 should include additional metrics like F1-score or precision/recall for a more comprehensive comparison.

8. Equation 6, Equation 7 are difficult to follow due to dense notation. Break them down into simpler steps or provide more explanatory text. Ensure all equations are referenced in the text. For example, Equation 10 (fitness function for GCO) is introduced without sufficient context. Explain how this function is used in the optimization process.

9. Many references are outdated (from 2017-2019). Include more recent studies (2022-2024) to reflect the latest advancements in fake news detection and unsupervised learning. Some references are incomplete or lack key details. For example, Reference does not include the conference name or page numbers. Ensure all references follow the required citation style. Ensure consistent formatting across references. For example, some references include full first names, while others use initials. Follow the journal's style guide.

Reviewer #7: In the manuscript, Dr. Perveen et al. proposed to use a hybrid method based on Gaussian Mixture Model (GMM) and the Group Counseling Optimizer (GCO) to detect fake news on social media. Their methodology are statistically sound and the results are promising. I have read their revised paper and their response to previous reviewer comments. They have addressed most of their concerns. Yet, there is one important comment that I found their answer unsatisfactory.

Both Reviewer #1 (Comment #3) and Reviewer #2 (Comment #7) suggested the authors to compare their method to some other available methods like the DBScan algorithms. But, the author only provided an empirical discussion about DBScan without showing any solid result ("Our prior experience with DBSCAN yielded unsatisfactory results in terms of clustering accuracy for fake news detection. Although these newer density-based algorithms (QuickDBScan,

KDTreeDBSCAN and DenLAC) may offer improvements, the reason to exclude them was likely based on the

failure of DBSCAN to meet the requirements for our specific dataset and the added complexity these

algorithms could introduce without guaranteed improvements.") I think the authors should further address this comment with some numerical results or real data analysis to show that why DBScan's results are unsatisfactory and the other available methods like QuickDBScan only provide limited improvement. This comment is important to help the readers to understand the advantages of their newly proposed method.

Apart from this comment, I also have a few Minor concerns:

Minor concerns:

1. How can users know which cluster is the fake news after applying your method? In your data analysis, you have the ground truth label. So, it's quite easy to find the cluster standing for fake news. But, in reality, people do not have access to such labels. Please discuss how to mitigate this problem.

2. The authors should double check the mathematical formulas in their manuscript. There are still many typos in the revised version:

(1) In line 228 and 230 on page 6, descriptive words in formula (2) and (3) should not be italic. But variables like t and n should be italic.

(2) In equation (5) on page 7, it should be \Lambda rather than \wedge.

(3) In line 275 on page 7, "D X D" should be $D \times D$.

(4) In equation 7 on page 8, \mathcal{N} is missing before (x|\mu_k,\Sigma_k). The same problem in line 281.

(5) In the sentence beginning from line 280, "Whereas \mu_k is the D-dimensional mean vector" should be ", where as \mu_k is the D-dimensional mean vector". Also, there is an unnecessary comma before \Sigma_k.

(6) In the denominator equation 8, there is an unnecessary word "that".

(7) In the end of line 312, a right parenthesis is missing.

**Do you want your identity to be public for this peer review?** For information about this choice, including consent withdrawal, please see our Privacy Policy

Reviewer #3: No

Reviewer #4: No

Reviewer #5: No

Reviewer #6: No

Reviewer #7: No

---

## [Author Response · Author response to Decision Letter 2]

24 Apr 2025

Respected Reviewers,

On behalf of my all co-authors, I am really grateful for your precious time to review our article and your inslightful comments that indeed improve the quality of our paper. In second round of revision, we have addressed your all constructive comments very carefully. The response against each comment is given in response to reviewer file.

The revised manuscript with highlighted text presents all the changes as per your kind directions.

Many Thanks.

Prfound Regard

Dr. M. Usman

---

## [Decision Letter · Decision Letter 2]

12 May 2025

Dear Dr. Ashraf,

Thank you for submitting your manuscript to PLOS ONE. After careful consideration, we feel that it has merit but does not fully meet PLOS ONE’s publication criteria as it currently stands. Therefore, we invite you to submit a revised version of the manuscript that addresses the points raised during the review process.

We look forward to receiving your revised manuscript.

Kind regards,

Fredrick Romanus Ishengoma

Academic Editor

PLOS ONE

Reviewers' comments:

Reviewer's Responses to Questions

**Comments to the Author**

Reviewer #6: All comments have been addressed

Reviewer #7: All comments have been addressed

2. Is the manuscript technically sound, and do the data support the conclusions?

Reviewer #6: Yes

Reviewer #7: (No Response)

3. Has the statistical analysis been performed appropriately and rigorously?

Reviewer #6: Yes

Reviewer #7: (No Response)

4. Have the authors made all data underlying the findings in their manuscript fully available?

Reviewer #6: Yes

Reviewer #7: (No Response)

5. Is the manuscript presented in an intelligible fashion and written in standard English?

Reviewer #6: Yes

Reviewer #7: (No Response)

Reviewer #6: 1. The citations in the paper are not presented in sequential order, which disrupts the flow and can confuse readers.

2. The references lack consistency in formatting. For example, reference [12] is incomplete and does not follow standard citation structure. Several entries are missing key details such as publication year or venue. A thorough revision is needed to ensure accuracy and adherence to a uniform citation style.

3. The paper presents the core GCO algorithm as a figure, which helps with visualization, but a textual pseudocode version would enhance clarity and reproducibility. Additionally, the algorithm's complexity and convergence behavior should be discussed to provide a deeper understanding of its scalability and efficiency.

4. There is a discrepancy between the table reference in the text ("Table 1") and its actual label ("Table 5"). This should be corrected to maintain proper cross-referencing.

5. Some sections, like the social media history in the introduction, are too long and could be shortened.

6. Equations need consistent formatting, and terms like "Silhouette Score" should be defined before use.

Reviewer #7: The authors have addressed all my concerns. Yet, I think they should add the response to my comment on "discussions about how users can decide which cluster is the fake news after applying their method" to their discussion part. It will be helpful for users to have some guidance on this point.

**Do you want your identity to be public for this peer review?** For information about this choice, including consent withdrawal, please see our Privacy Policy

Reviewer #6: No

Reviewer #7: No

---

## [Author Response · Author response to Decision Letter 3]

20 May 2025

Respected Reviewers,

On behalf of all co-authors, I am really grateful to all the reviewers for their precious time to review and giving constructive comments.

We are uploading (a) our point-by-point response to the comments (below) (response to reviewers, under “Response-to-Reviewers Files”), (b) an updated manuscript with yellow highlighting indicating changes (as “Revised Article with Changes Highlighted”), and (c) a clean updated manuscript without highlights (“Manuscript”).

thanks.

---

## [Decision Letter · Decision Letter 3]

1 Aug 2025

Unsupervised fake news detection on social media using hybrid gaussian mixture model

PONE-D-24-50144R3

Dear Dr. Ashraf,

We’re pleased to inform you that your manuscript has been judged scientifically suitable for publication and will be formally accepted for publication once it meets all outstanding technical requirements.

Kind regards,

Hirenkumar Kantilal Mewada

Academic Editor

PLOS ONE

Additional Editor Comments (optional):

Reviewers' comments:

Reviewer's Responses to Questions

**Comments to the Author**

Reviewer #6: All comments have been addressed

Reviewer #7: All comments have been addressed

2. Is the manuscript technically sound, and do the data support the conclusions?

Reviewer #6: Yes

Reviewer #7: Yes

3. Has the statistical analysis been performed appropriately and rigorously?

Reviewer #6: Yes

Reviewer #7: Yes

4. Have the authors made all data underlying the findings in their manuscript fully available?

Reviewer #6: Yes

Reviewer #7: Yes

5. Is the manuscript presented in an intelligible fashion and written in standard English?

Reviewer #6: Yes

Reviewer #7: Yes

Reviewer #6: I have reviewed the revised manuscript and confirm that all previous comments have been addressed:

Citations are now sequential; The introduction is more concise now and clearly explains dynamic systems and GMMs; The algorithm includes textual pseudocode with complexity and convergence discussion; Table references are corrected; Equations are consistently formatted, and terms like “Silhouette Score” are defined.

Overall, the manuscript has improved in clarity and presentation.

Reviewer #7: (No Response)

**Do you want your identity to be public for this peer review?** For information about this choice, including consent withdrawal, please see our Privacy Policy

Reviewer #6: No

Reviewer #7: No

---

## [Editor Report · Acceptance letter]

PONE-D-24-50144R3

PLOS ONE

Dear Dr. Ashraf,

I'm pleased to inform you that your manuscript has been deemed suitable for publication in PLOS ONE. Congratulations! Your manuscript is now being handed over to our production team.

Kind regards,

on behalf of

Dr. Hirenkumar Kantilal Mewada

Academic Editor

PLOS ONE